# A bacterial gene acquired by parasitoid wasps contributes to venom secretion against host defence

Zhiguo Liu[1,2,5], Mei Tao[1,2,5], Zixuan Xu [ID][1,2,5], Junwei Zhang[1,2,5], Yang Li [ID][1,2], Zhi Dong [ID][1,2], Qichao Zhang[1,2], Lan Pang[1,2], Yifeng Sheng[1,2], Yueqi Lu[1,2], Ting Feng[1,2], Wenqi Shi[1,2], Longtao Yu[1,2], Antonis Rokas [ID][3], Jiani Chen [ID][1,2✉], Xing-Xing Shen [ID][1,2,4✉] & Jianhua Huang [ID][1,2✉]

## Abstract

**Horizontal gene transfer (HGT) is an important source of gene innovation in prokaryotic and eukaryotic organisms. Several genes acquired by hosts of parasitoid wasps via HGT have been reported to protect hosts from parasitoid wasps. In contrast, little is known about whether HGT-acquired genes in parasitoid wasps are involved in attacking their hosts. Here, we report a prokaryote-type CDP-diacylglycerol synthase (*PTCDS*) gene that was horizontally transferred into the last common ancestor of two parasitoid wasps, *Leptopilina heterotoma* and *L. syphax*, from the bacterial family Rickettsiaceae. We experimentally demonstrated that *PTCDS* is linked to ensure the appropriate storage amount of venom in the venom reservoir of parasitoid wasps. *PTCDS* knockdown downregulated the expression of certain vesicle-mediated transport genes, thereby reducing the secretion of venom into venom reservoir without altering its composition. This resulted in a significant increase in the proportion of encapsulated wasp eggs in parasitized hosts, ultimately leading to host immune-mediated killing. We conclude that parasitoid wasps use the foreign gene *PTCDS* to influence venom amounts against host defence, providing new insight into the arms race between parasitoid wasps and hosts.**

**Keywords** HGT; Parasitoid–Host Competition; Venom Proteins; Evolutionary Genomics; Adaptive Evolution
**Subject Category** Evolution & Ecology

## Introduction

Horizontal gene transfer (HGT), the transmission of genetic material between distantly related organisms by non-reproductive means, plays an important role in genome evolution (Husnik and McCutcheon, 2018; Irwin et al, 2022; Perreau and Moran, 2022). HGT is rampant in prokaryotes, but comparative genomic analyses have substantially changed our understanding of HGT in eukaryotes (Dunning Hotopp, 2011; Keeling and Palmer, 2008; Kominek et al, 2019; Ochman et al, 2000). For example, in insects, a lineage containing >50% of all animals, many HGT-acquired genes have been reported to contribute to important traits (Dai et al, 2021; Di Lelio et al, 2019; Gasmi et al, 2021; Gilbert and Maumus, 2023; Liu et al, 2023; Meng et al, 2009; Moran and Jarvik, 2010; Parker and Brisson, 2019; Verster et al, 2023; Walker et al, 2023; Xia et al, 2021; Wang et al, 2023; Li et al, 2024), such as body colouration, detoxification, immune protection, and sex determination. In addition, a recent large-scale, systematic survey of 218 insect genomes revealed 1410 foreign genes from non-metazoan (mostly bacterial) donors via HGT and identified a foreign gene that contributes to male courtship in lepidopterans (Li et al, 2022).

Parasitoid wasps (Hymenoptera) constitute one of the most fascinating groups of insects (Huang et al, 2026; Fei et al, 2023; Stork, 2018). Unlike common venomous organisms (e.g., snakes, spiders, and scorpions), which primarily use venom for prey and/or defence, parasitoid wasps use venom for offspring reproduction in or on their hosts (mostly other insects) (Mrinalini and Werren, 2017). In particular, parasitoid wasps use their venom to alter the immunity, growth, and metabolism of their hosts so that their eggs can hatch successfully, and the resulting juvenile wasp larvae can harvest enough nutritional supplies from the bodies of their hosts (Asgari and Rivers, 2011; Moreau and Asgari, 2015; Poirié et al, 2014). In response, hosts usually strengthen their immune systems (e.g., haemocytes) (Abram et al, 2019; Blumberg, 1997) and/or utilize defensive symbionts (Hansen et al, 2012; Oliver et al, 2009; Vorburger, 2022) to suppress parasitoid wasp parasitism.

Previous studies have experimentally shown that several HGT-acquired genes contribute to host defences against parasitoid wasp parasitism (Di Lelio et al, 2019; Gasmi et al, 2021; Verster et al, 2023). Specifically, a gasmin gene transferred from a symbiotic virus to cotton leafworms, contributes to the immune defence barrier against parasitoid wasp invasion (Di Lelio et al, 2019), a parasitoid killing factor gene transferred from a virus to lepidopterans contributes to the killing capacity against parasitoid wasp parasitism (Gasmi et al, 2021), and two toxic genes transferred from bacteria to fruit flies contribute to protection

[1]Zhejiang Key Laboratory of Biology and Ecological Regulation of Crop Pathogens and Insects, Institute of Insect Sciences, College of Agriculture and Biotechnology, Zhejiang University, 310058 Hangzhou, China. [2]Ministry of Agriculture Key Laboratory of Molecular Biology of Crop Pathogens and Insects, Zhejiang University, 310058 Hangzhou, China. [3]Department of Biological Sciences and Evolutionary Studies Initiative, Vanderbilt University, Nashville, TN 37235, USA. [4]Centre for Evolutionary & Organismal Biology, Zhejiang University, 310058 Hangzhou, China. [5]These authors contributed equally: Zhiguo Liu, Mei Tao, Zixuan Xu, Junwei Zhang. ✉E-mail: jnchen@zju.edu.cn; xingxingshen@zju.edu.cn; jhhuang@zju.edu.cn

from parasitoid wasp attacks (Verster et al, 2023). However, little information is currently available concerning that parasitoid wasps also contain HGT-acquired genes from bacteria, fungi, and viruses. Most importantly, their roles in parasitoid wasp parasitism remain largely unknown because of a lack of in vivo experimental evidence.

We undertook a systematic investigation of HGT-acquired genes in the genome of the *Leptopilina* parasitoid wasps (*L. heterotoma*, *L. syphax*, *L. drosophilae*, *L. boulardi*, and *L. myrica*) that attack *Drosophila* hosts via a robust and conservative phylogeny-based approach. We identified 19 HGT-acquired genes into the five genomes of parasitoid wasps, including an uncharacterized foreign gene in insects, namely, a prokaryote-type CDP-diacylglycerol synthase (*PTCDS*) gene. *PTCDS* was horizontally acquired by the last common ancestor of *L. heterotoma* and *L. syphax* from a donor in the bacterial family Rickettsiaceae. We further provide experimental evidence that *PTCDS* governs the amount of venom used against host immune defence, ensuring the successful growth and development of wasp offspring within *Drosophila* hosts.

# Results

## Systematic examination of HGT-acquired genes in five parasitoid wasps of the genus *Leptopilina*

To systematically identify putative HGT-acquired genes in the *Leptopilina* parasitoid wasps, we first retrieved five (*L. heterotoma*, *L. syphax*, *L. drosophilae*, *L. boulardi*, and *L. myrica*) publicly available genomes and their gene annotations from GenBank data via FTP (Dataset EV1). Next, we used a robust and conservative phylogeny-based approach (Li et al, 2022) to identify putative HGT-acquired genes. We identified a total of 19 genes in the five wasp genomes that were acquired via five distinct transfer events from bacteria (Fig. 1A; Dataset EV2). Among the five distinct events, three HGT events occurred in the common ancestor and two events were species-specific (Fig. 1B; Dataset EV2).

To investigate the expression profiles of all 19 HGT-acquired genes, we generated transcriptome data for seven different tissues, including the antennae, head, thorax, abdomen, leg, ovary, and venom gland, from adult females of each parasitoid wasp species (*L. heterotoma*, *L. syphax*, *L. drosophilae*, *L. boulardi*, and *L. myrica*) (Fig. 1C). By comparing the expression levels of the HGT-acquired genes across tissues and species, we found that the HGT-acquired gene *LhChr005.253* and its ortholog *LsChr004.1071*, which correspond to HGT event3 (Fig. 1B), presented the highest expression levels in the venom gland (Fig. 1D). The gene family phylogeny revealed that the HGT-acquired genes *LhChr005.253* and *LsChr004.1071* were acquired by the last common ancestor of *L. heterotoma* and *L. syphax* from a donor gene in the bacterial family Rickettsiaceae (Figs. 1E; EV1) and was predicted to be the prokaryote-type CDP-diacylglycerol synthase (henceforth termed the *PTCDS* gene). Furthermore, the analysis of gene orders between the genome assemblies of five parasitoid wasps revealed that the orders of the foreign gene *PTCDS* and its surrounding native genes were highly conserved across all five wasps, except for *L. boulardi* (Fig. 1F). Collectively, these results suggest that the foreign wasp gene *PTCDS* might play an important role in the venom gland, which is crucial for parasitoid wasp parasitism.

## Knockdown of horizontally acquired *PTCDS* reduces the emergence rate of wasp offspring

To examine the function of the HGT-acquired gene *PTCDS* in parasitoid wasps, we used RNAi to knock down the expression level of *PTCDS* in females from *L. heterotoma* and *L. syphax* (see Methods). qRT-PCR validation revealed a highly efficient knockdown of *PTCDS*, with an efficiency of 89% ± 2.5% in *L. heterotoma* and 81% ± 1.3% in *L. syphax*. Both results were statistically significant compared to their *dsGFP* controls (*L. heterotoma*: $P = 6.5 \times 10^{-5}$, $n = 3$; *L. syphax*: $P = 3.0 \times 10^{-4}$, $n = 3$; two-tailed Student's *t* test; Fig. EV2). Compared with the control female adult wasps, the knockdown female adult wasps presented no significant differences in body size (*L. heterotoma*: $P = 0.15$, $n = 30$; *L. syphax*: $P = 0.64$, $n = 25$; two-tailed Student's *t* test; Fig. EV3A; Dataset EV3), ovary size (*L. heterotoma* ovary length: $P = 0.45$, $n = 30$; *L. heterotoma* ovary width: $P = 0.14$, $n = 30$; *L. syphax* ovary length: $P = 0.12$, $n = 25$; *L. syphax* ovary width: $P = 0.23$, $n = 25$; two-tailed Student's *t* test; Fig. EV3B; Dataset EV3), or host-searching behaviours, including both the propensity for host-searching (*L. heterotoma*: $P = 0.69$, $n = 6$; *L. syphax*: $P = 0.71$, $n = 6$; two-tailed Student's *t* test) and the time spent searching (*L. heterotoma*: $P = 0.29$, $n = 15$; *L. syphax*: $P = 0.67$, $n = 15$; two-tailed Student's *t* test; Fig. EV4A,B; Dataset EV4).

Next, we examined the oviposition rate (measured by the proportion of host larvae parasitized by wasps) and the emergence rate (measured by the proportion of wasp offspring that successfully emerged from the host larvae) of the knockdown and control female adult wasps on each of 6 different *Drosophila* species hosts (*D. melanogaster*, *D. simulans*, *D. sechellia*, *D. pseudoobscura*, *D. mauritiana*, and *D. santomea*). We found that the *PTCDS*-knockdown wasps presented similar oviposition rates across all 6 host species (*D. melanogaster*: $P = 0.52$ for *L. heterotoma*, $P = 0.89$ for *L. syphax*, $n = 3$; *D. simulans*: $P = 0.72$ for *L. heterotoma*, $P = 0.63$ for *L. syphax*, $n = 3$; *D. sechellia*: $P = 0.39$ for *L. heterotoma*, $P = 0.49$ for *L. syphax*, $n = 3$; *D. pseudoobscura*: $P = 0.83$ for *L. heterotoma*, $P = 0.62$ for *L. syphax*, $n = 3$; *D. mauritiana*: $P = 0.08$ for *L. heterotoma*, $P = 0.34$ for *L. syphax*, $n = 3$; *D. santomea*: $P = 0.85$ for *L. heterotoma*, $P = 0.61$ for *L. syphax*, $n = 3$; two-tailed Student's *t* test; Fig. EV5; Dataset EV5) but had significantly lower emergence rates than the *dsGFP* control wasps (*D. melanogaster*: $P = 2.4 \times 10^{-5}$ for *L. heterotoma*, $P = 3.3 \times 10^{-4}$ for *L. syphax*, $n = 3$; *D. simulans*: $P = 1.1 \times 10^{-3}$ for *L. heterotoma*, $P = 1.1 \times 10^{-2}$ for *L. syphax*, $n = 3$; *D. sechellia*: $P = 1.8 \times 10^{-3}$ for *L. heterotoma*, $P = 1.1 \times 10^{-2}$ for *L. syphax*, $n = 3$; *D. pseudoobscura*: $P = 4.2 \times 10^{-4}$ for *L. heterotoma*, $P = 1.5 \times 10^{-2}$ for *L. syphax*, $n = 3$; *D. mauritiana*: $P = 5.6 \times 10^{-3}$ for *L. heterotoma*, $P = 5.1 \times 10^{-4}$ for *L. syphax*, $n = 3$; *D. santomea*: $P = 8.1 \times 10^{-4}$ for *L. heterotoma*, $P = 1.2 \times 10^{-3}$ for *L. syphax*, $n = 3$; two-tailed Student's *t* test; Fig. 2A; Dataset EV5). Specifically, the emergence rates varied in the different host species between $0.04 \pm 0.02$ and $0.46 \pm 0.05$, with an average value of $0.31 \pm 0.03$ for the knockdown *L. heterotoma* but between $0.53 \pm 0.05$ and $0.82 \pm 0.02$, with an average value of $0.72 \pm 0.03$ for the control *L. heterotoma* (Fig. 2A). The emergence rates also varied in the different host species between $0.12 \pm 0.03$ and $0.23 \pm 0.04$, with an average value of $0.17 \pm 0.02$ for the knockdown *L. syphax* but between $0.42 \pm 0.02$ and $0.59 \pm 0.01$, with an average value of $0.50 \pm 0.02$ for the control *L. syphax* (Fig. 2A).

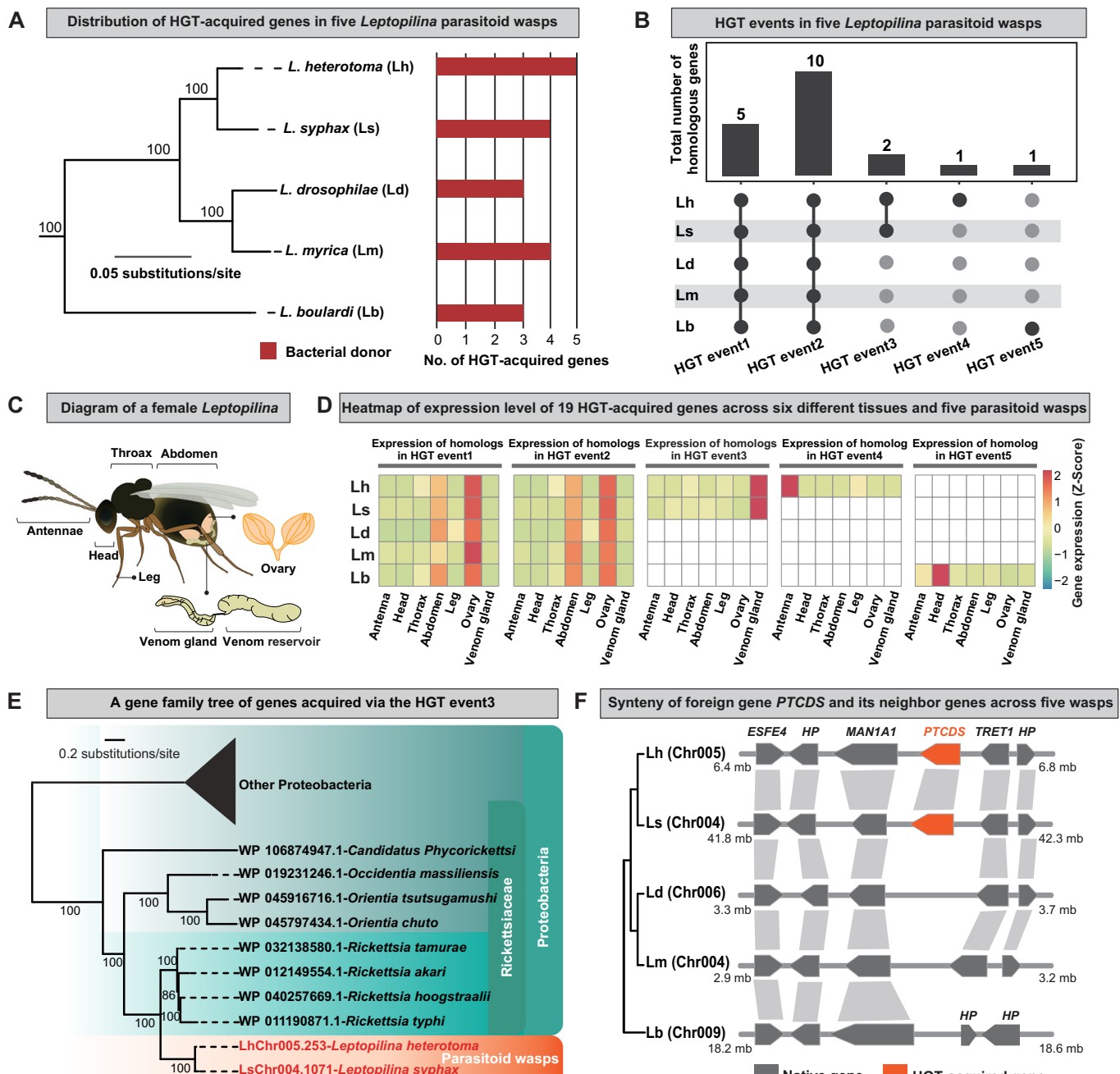

**A** Distribution of HGT-acquired genes in five *Leptopilina* parasitoid wasps

**B** HGT events in five *Leptopilina* parasitoid wasps

**C** Diagram of a female *Leptopilina*

**D** Heatmap of expression level of 19 HGT-acquired genes across six different tissues and five parasitoid wasps

**E** A gene family tree of genes acquired via the HGT event3

**F** Synteny of foreign gene *PTCDS* and its neighbor genes across five wasps

To identify the underlying causes of the lower emergence rates of the *PTCDS*-knockdown wasps, we first examined the encapsulation rate of the *D. melanogaster* host larvae parasitized by the knockdown and control female adult wasps. We found that the encapsulation rate of the host larvae parasitized by the *PTCDS*-knockdown wasps (knockdown *L. heterotoma*: 49% ± 1.3%; knockdown *L. syphax*: 60% ± 4.6%) was significantly greater than that of host larvae parasitized by the control wasps (control *L. heterotoma*: 16% ± 3.7%; control *L. syphax*: 2% ± 0.4%) (*L. heterotoma*: $P = 4.4 \times 10^{-6}$, $n = 3$; *L. syphax*: $P = 1.8 \times 10^{-3}$, $n = 3$; two-tailed Student's *t* test; Fig. 2B; Dataset EV6). Because the encapsulation phenotype arises from the host immune defence that typically

mobilizes host haemocytes (blood cells) to kill wasp eggs (Colinet et al, 2013; Colinet et al, 2007; Huang et al, 2021; Mortimer, 2013; Stephenson et al, 2022), we next assessed the number of haemocytes in the *D. melanogaster* host larvae parasitized by the *PTCDS*-knockdown and control female adult wasps. We found that the number of haemocytes in the host larvae parasitized by the *PTCDS*-knockdown wasps was significantly greater (average number of haemocytes per host larva, knockdown *L. heterotoma*: 5330 ± 226; knockdown *L. syphax*: 5760 ± 159) than that of the haemocytes in the host larvae parasitized by the control wasps (the average number of haemocytes per host larva, control *L. heterotoma*: 3380 ± 187; control *L. syphax*: 3630 ± 304)

**Figure 1.   Nineteen parasitoid wasp genes were acquired via horizontal gene transfer (HGT).**

(**A**) Distribution of 19 HGT-acquired genes on the concatenated maximum likelihood (ML) tree of five *Leptopilina* parasitoid wasps inferred from analysis of 1,367 BUSCO genes under a single GTR + G4 + F using IQ-TREE multicore version 1.6.8 (Nguyen et al, 2015). Branch support values near internodes correspond to ultrafast bootstrap support. Two outgroups (*Asobara japonica* and *Diachasma alloeum*) are not shown. Note that the donors of all 19 putative HGT-acquired genes are bacterial (in red bar). Detailed information of putative HGT donors is given in Dataset EV2. (**B**) Summary of five distinct HGT events. These 19 genes were acquired through five distinct HGT events, of which three events involved two or more species and two were species-specific. The gray bar indicates the number of homologous genes found in each of five distinct HGT events. Among five distinct HGT events, the HGT event2 contains multiple-copy homologs for *L. heterotoma* (two copies), *L. syphax* (two copies), *L. drosophilae* (two copies) and *L. myrica* (three copies) and a single-copy homolog for *L. boulardi*; the remaining four HGT events contain only one single-copy homolog for each recipient wasp. (**C**) A simplified diagram of a wild-type female adult parasitoid wasp, in which seven different tissues including antennae, head, thorax, abdomen, leg, ovary and venom gland, are indicated. (**D**) Quantification of the relative gene expression levels of all 19 foreign genes acquired via five distinct HGT events, using transcriptome data of seven different tissues from the wild-type female adult of each of five parasitoid wasps. The cell boxes with filled colors denote transformed relative expression level of the examined HGT-acquired gene, while cell boxes without filled colors denote the absence of the examined HGT-acquired gene. Note that we averaged the expression levels of the examined HGT-acquired genes with multiple-copy homologs in the HGT event2. Clearly, the foreign genes *LhChr005.253* and *LsChr004.1017*, which correspond to HGT event3, had the highest expression levels in the venom gland. (**E**) A simplified gene family phylogeny of the foreign genes *LhChr005.253* and *LsChr004.1017* that correspond to HGT event3. Taxa in red indicate parasitoid wasps *L. heterotoma* and *L. syphax*, while taxa in black indicate Bacteria. This result showed that the foreign genes *LhChr005.253* and *LsChr004.1017* were horizontally acquired by the last common ancestor of *L. heterotoma* and *L. syphax* from a donor the bacterial family Rickettsiaceae. The full phylogenetic tree for *LhChr005.253* and *LsChr004.1017* is provided in Fig. EV1. (**F**) Synteny of gene orders in the genomic region that harbors the foreign genes *LhChr005.253* and *LsChr004.1017* across five parasitoid wasps. The HGT-acquired genes *LhChr005.253* and *LsChr004.1017* were predicted as the prokaryote-type CDP-diacylglycerol synthase in *Escherichia coli*. Here, we named the foreign genes *LhChr005.253* and *LsChr004.1017* as *PTCDS*. ESFE4 esterase FE4-like, HP hypothetical protein, MAN1A1 mannosidase alpha class 1A member 1, TRET1 facilitated trehalose transporter Tret1.

(*L. heterotoma*: $P = 3.1 \times 10^{-6}$, $n = 10$; *L. syphax*: $P = 7.4 \times 10^{-6}$, $n = 10$; two-tailed Student's *t* test; Fig. 2C; Dataset EV6). Collectively, these results show that the knockdown of *PTCDS* in parasitoid wasps leads to a lower emergence rate of wasp offspring from host larvae because of the lower efficiency of suppressing host cellular immune defence.

## Knockdown of the horizontally acquired *PTCDS* reduces the total venom protein amount but not its composition in the venom reservoir

Since the parasitoid venom gland (which synthesizes and secretes venom proteins) and the venom reservoir (which stores venom proteins) usually give rise to venom proteins that suppress host immune defence (Mrinalini and Werren, 2017), we examined their developmental phenotypes in *PTCDS*-knockdown and control female adult wasps. We found that the knockdown wasps had no significant differences in venom gland size (average venom gland volume, knockdown *L. heterotoma*: $0.0039 \pm 0.0002$ mm³; knockdown *L. syphax*: $0.0014 \pm 0.0001$ mm³) compared with the control wasps (average venom gland volume, control *L. heterotoma*: $0.0043 \pm 0.0002$ mm³; control *L. syphax*: $0.0013 \pm 0.0001$ mm³) (*L. heterotoma*: $P = 0.21$, $n = 30$; *L. syphax*: $P = 0.28$, n = 25; Mann–Whitney *U* test for *L. heterotoma*, two-tailed Student's *t* test for *L. syphax*; Fig. 3A; Dataset EV7). Surprisingly, when we compared the venom reservoir size between the *PTCDS*-knockdown and control female adult wasps, we found that the venom reservoir size of the knockdown wasps (average venom reservoir volume, knockdown *L. heterotoma*: $0.0014 \pm 0.0001$ mm³; knockdown *L. syphax*: $0.0009 \pm 0.0001$ mm³) was ~threefold to fourfold smaller than that of the control wasps (average venom reservoir volume, control *L. heterotoma*: $0.0056$ mm³ $\pm 0.0003$; control *L. syphax*: $0.0023 \pm 0.0001$ mm³) (*L. heterotoma*: $P = 3.5 \times 10^{-16}$, $n = 30$; *L. syphax*: $P = 5.6 \times 10^{-10}$, $n = 25$; Mann–Whitney *U* test; Fig. 3A; Dataset EV7).

Given the substantially reduced size of the venom reservoir in the *PTCDS*-knockdown female adult wasps, we next investigated total venom protein amount from the venom reservoir between *PTCDS*-knockdown and control female adult *L. heterotoma* wasps via SDS-PAGE, quantitative protein assay kit, and mass spectrometry (LC-MS/MS). The SDS-PAGE gel electrophoretic profile revealed that the total venom protein amount in venom reservoir was visibly lower for the *PTCDS*-knockdown *L. heterotoma* than for the control *L. heterotoma* (left panel in Fig. 3B). Consistent with the results of SDS-PAGE, quantification of the amount of total venom proteins via the protein assay kit revealed that the venom reservoir had, on average, a 2-fold lower amount of total venom proteins for the *PTCDS*-knockdown *L. heterotoma* ($1.4 \pm 0.1$ µg per venom reservoir) than for the control *L. heterotoma* ($2.8 \pm 0.1$ µg per venom reservoir) (middle panel in Fig. 3B; Dataset EV8). Moreover, to precisely investigate the compositions of venom proteins in the venom reservoir, we generated proteome data of venom proteins via LC-MS/MS and then identified venom proteins based on an integrated transcriptomic and proteomic approach (Huang et al, 2021; Martinson et al, 2017). We found that the compositions of venom proteins in the venom reservoir of the *PTCDS*-knockdown wasps were nearly identical to those in the venom reservoir of the control wasps (right panel in Fig. 3B; Dataset EV9). Notably, PTCDS itself was not detected among the identified venom proteins (Dataset EV9).

Finally, western blot analysis of the venom protein marker Lar (lymph gland apoptosis-related protein) reported in a previous study (Huang et al, 2021) revealed that the amount of Lar was not significantly different in the venom glands of the *PTCDS*-knockdown vs. the control wasps but was ~twofold lower in the venom reservoir of the *PTCDS*-knockdown wasps than in the venom reservoir of the control wasps (Fig. 3C; Dataset EV10). Overall, these results show that knockdown of the HGT-acquired gene *PTCDS* reduced the total amount but did not change the composition of venom proteins in the venom reservoir. This suggests a link between PTCDS function and the regulation of stored venom protein amount, which could be mediated by affecting protein secretion in the venom gland cells or by influencing venom reservoir size.

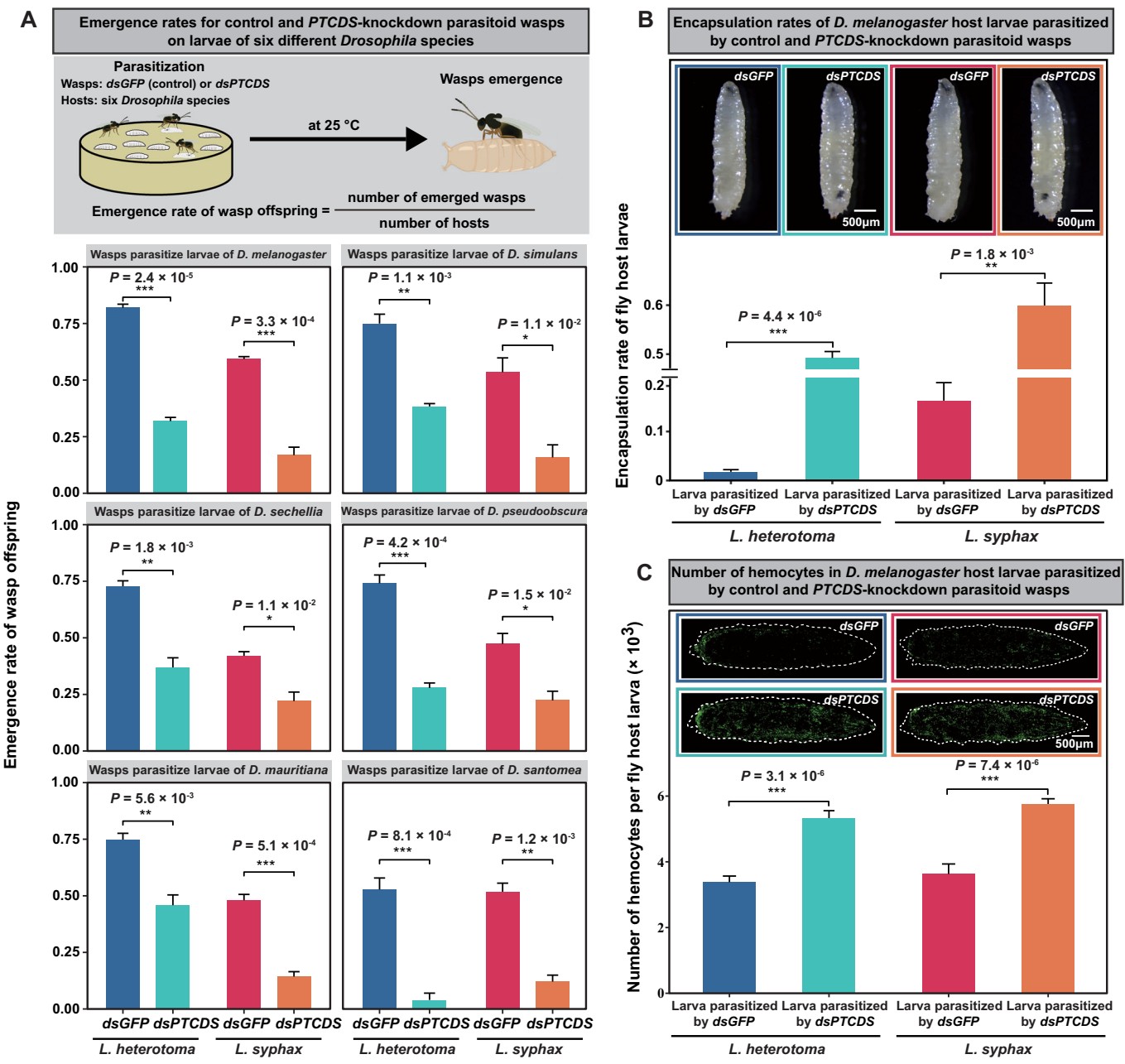

**Figure 2. *PTCDS*-defective parasitoid wasps presented a substantially lower emergence rate of their offspring.**

(**A**) Effect of *PTCDS* on the emergence rate of wasp offspring from the six different *Drosophila* host larvae. We used RNAi to knock down the expression level of *PTCDS* in *L. heterotoma* and *L. syphax* (see "Methods"). *dsGFP* was used as a control. The ratios of female wasps to host larvae were 3:360 and 3:180 for *L. heterotoma* and *L. syphax*, respectively. Three-day-old female adult *L. heterotoma* and 9-day-old female adult *L. syphax* were used to parasitize the host larvae separately. Three replicates were included for each treatment. Data are presented as mean ± SEM. Significance was determined by two-tailed Student's *t* test (*P < 0.05; **P < 0.01; ***P < 0.001). (**B**) Effect of *PTCDS* on the encapsulation rate of *D. melanogaster* host larvae parasitized by 3-day-old female adult *L. heterotoma* and 9-day-old female adult *L. syphax*. *dsGFP* was used as a control. Three replicates were included for each treatment. Data are presented as mean ± SEM. Significance was determined by two-tailed Student's *t* test (**P < 0.01; ***P < 0.001). (**C**) Effect of *PTCDS* on the number of haemocytes (blood cells) in *D. melanogaster* host larvae parasitized by 3-day-old female adult *L. heterotoma* and 9-day-old female adult *L. syphax*. Note that *Hml-GAL4 > UAS-GFP* was a haemocyte-specific *GAL4* and GFP fluorescent reporter fusion line, in which all circulating haemocytes were detected via a GFP reporter. Ten replicates were included for each treatment. Data are presented as mean ± SEM. Significance was determined by two-tailed Student's *t* test (***P < 0.001). Source data are available online for this figure.

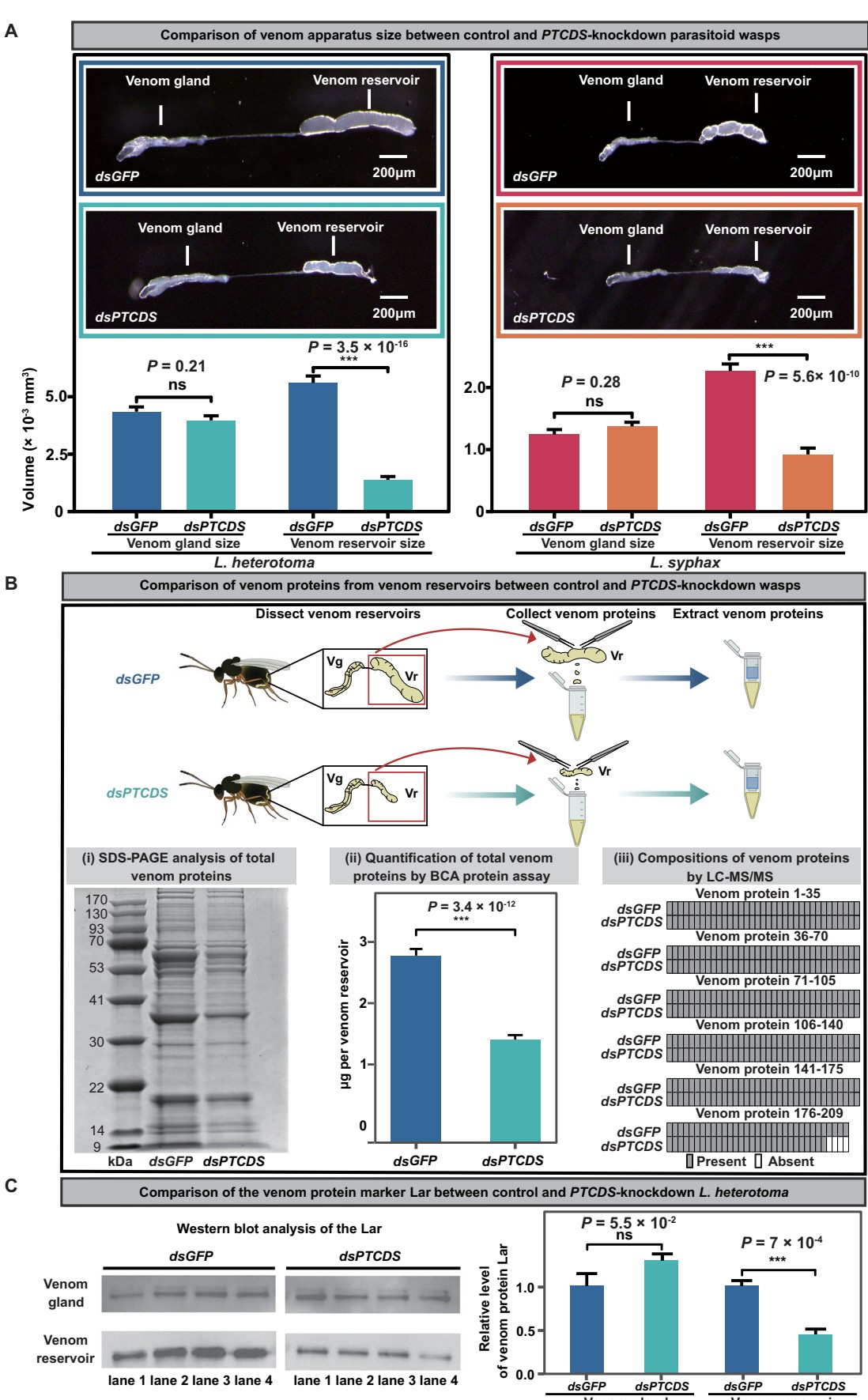

**Figure 3.    *PTCDS* influences the total venom protein amount but does not affect its composition in the venom reservoir.**

(A) Effects of *PTCDS* on venom gland size and venom reservoir size in the knockdown and control parasitoid wasps, respectively. Three-day-old female adult *L. heterotoma* (*n* = 30) and 9-day-old female adult *L. syphax* (*n* = 25) were used to examine their venom gland sizes and venom reservoir sizes, respectively. *dsGFP* was used as a control. Data are presented as mean ± SEM. Statistical analysis was performed using two-tailed Student's *t* test when parametric assumptions and homogeneity of variances were met, and the Mann–Whitney *U* test was used to determine significance when experiments required nonparametric statistical tests (ns, not significant; ***P < 0.001). (B) Effects of *PTCDS* on the total amount and composition of venom proteins in the venom reservoirs of the knockdown and control *L. heterotoma* parasitoid wasp. *dsGFP* was used as a control. The total amount of venom proteins was assessed via SDS-PAGE (i) and a quantitative protein assay kit (ii). The compositions of venom proteins (iii) were determined via LC-MS/MS on previously reported venom proteins in *L. heterotoma*. These results show that the knockdown of *PTCDS* reduced the amount of total venom proteins, while it barely changed the composition of venom proteins. Venom proteins from 20 venom reservoirs were loaded in each lane for SDS-PAGE, and venom proteins per venom reservoir were measured via BCA protein assay (*n* = 30). (C) Comparison of the venom protein marker Lar between the venom gland and venom reservoir in 3-day-old control and *PTCDS*-knockdown *L. heterotoma*. The venom protein Lar (lymph gland apoptosis-related protein) was previously reported in *L. heterotoma* (Huang et al, 2021). The western blots were stained with the Lar-specific antibody for the venom gland and venom reservoir in the control (*dsGFP*) and knockdown (*dsPTCDS*) *L. heterotoma*. Venom proteins from ten venom glands and one venom reservoir were loaded in each lane. The relative concentration of Lar in the *PTCDS*-knockdown wasps was normalized to that in the control wasps for the venom gland and venom reservoir (*n* = 4). Data are presented as mean ± SEM. Significance was determined by two-tailed Student's *t* test (ns, not significant; ***P < 0.001). Source data are available online for this figure.

## The horizontally acquired *PTCDS* likely regulates venom protein secretion in venom gland cells

The question then arises of how the knockdown of *PTCDS* affects the amount of total venom proteins. To address this, we first quantified the expression levels of the foreign gene *PTCDS* in five abdomen tissues, including the ovary, fat body, gut, venom gland, and venom reservoir, from wild-type female adult *L. heterotoma* wasps. The qRT-PCR results revealed that *PTCDS* expression was highly enriched in the venom gland, with transcript levels ~3750-, 24-, 24-, and 9-fold higher than those in the ovary, fat body, gut, and venom reservoir, respectively (Fig. 4A). Next, we generated transcriptome data for the venom glands of *PTCDS*-knockdown and control female adult *L. heterotoma*. Functional enrichment analysis of differentially expressed genes (DEGs) between the *PTCDS*-knockdown and control *L. heterotoma* revealed that 446 significantly downregulated genes were involved mainly in lipid metabolic process, intracellular transport, vesicle-mediated transport, homeostatic process, protein maturation, response to abiotic stimulus, glycoprotein metabolic process, organophosphate biosynthetic process, and endomembrane system organization (Fig. 4B; Dataset EV11). This enrichment suggests that the foreign gene *PTCDS*, in addition to its possible role in lipid biosynthesis reported in previous *E. coli* studies (Icho et al, 1985; Sawasato et al, 2019), might also be involved in vesicle-associated biological processes. We selected 11 genes shared between the "intracellular transport" and "vesicle-mediated transport" categories for functional investigation: *Sar1*, *Vps53*, *CG10524*, *p24-1*, *Sec24AB*, *AP-1μ*, *Snx17*, *AP-1γ*, *CHOp24*, *CG1116*, and *CG5510* (Fig. 4C; Dataset EV12). We successfully generated dsRNA targeting eight of these genes (*Sar1*, *Vps53*, *p24-1*, *Sec24AB*, *AP-1μ*, *CHOp24*, *CG1116*, and *CG5510*) for RNAi knockdown experiments in female *L. heterotoma* wasps, while dsRNA synthesis failed for the remaining three genes (*CG10524*, *Snx17*, and *AP-1γ*) (see "Methods"). qRT-PCR analysis confirmed highly efficient gene knockdown (Fig. EV6). Notably, *L. heterotoma* wasps with gene knockdowns showed no significant differences in venom gland size compared to controls, with the exception of *CG1116* knockdown, which resulted in enlarged venom glands (*dsSar1*: *P* = 0.35, *n* = 30; *dsVps53*: *P* = 0.21, *n* = 30; *dsp24-1*: *P* = 0.99, *n* = 30; *dsSec24AB*: *P* = 0.32, *n* = 30; *dsAP-1μ*: *P* = 0.91, *n* = 30; *dsCHOp24*: *P* = 0.99, *n* = 30; *dsCG1116*: *P* = 3.0 × 10$^{-7}$, *n* = 30; *dsCG5510*: *P* = 0.99, *n* = 30;

Kruskal–Wallis test with Dunn's multiple comparison test; Fig. 4D; Dataset EV13). Furthermore, knockdown of *p24-1*, *Sec24AB*, *AP-1μ*, *CHOp24*, and *CG1116* significantly reduced venom reservoir size relative to controls, whereas no such effect was observed for *Sar1*, *Vps53*, or *CG5510* knockdowns (*dsSar1*: *P* = 0.99, *n* = 30; *dsVps53*: *P* = 0.99, *n* = 30; *dsp24-1*: *P* = 2.0 × 10$^{-9}$, *n* = 30; *dsSec24AB*: *P* = 3.7 × 10$^{-4}$, *n* = 30; *dsAP-1μ*: *P* = 6.0 × 10$^{-11}$, *n* = 30; *dsCHOp24*: *P* = 1.9 × 10$^{-10}$, *n* = 30; *dsCG1116*: *P* = 1.0 × 10$^{-15}$, *n* = 30; *dsCG5510*: *P* = 0.99, *n* = 30; Kruskal–Wallis test with Dunn's multiple comparison test; Fig. 4D; Dataset EV13). In contrast, the functional enrichment analysis of the 523 significantly upregulated genes revealed that these genes are involved mainly in ribosome biogenesis, cytoplasmic translation, and RNA modification (Fig. EV7). These results show that *PTCDS* might regulate venom secretion into the venom reservoir by modulating some protein transport-related genes in venom gland, including *p24-1*, *Sec24AB*, *AP-1μ*, *CHOp24*, and *CG1116*.

Indeed, venom protein secretion depends on a vesicle-associated transport system. Previous studies have shown that venom proteins are delivered by intracellular vesicles (IVs) from the extensive rough endoplasmic reticulum (ER) and Golgi apparatus to the microvillar region of rough canals in the venom gland secretory cells of *Leptopilina* wasps (Chiu et al, 2006; Ferrarese et al, 2009; Gueguen et al, 2011; Morales et al, 2005). Then, venom proteins are reassembled and reloaded into specialized extracellular microvesicles, which are now termed mixed-strategy extracellular vesicles (MSEVs) and were previously known as virus-like particles (Ferrarese et al, 2009; Gueguen et al, 2011; Morales et al, 2005; Heavner et al, 2017). Finally, MSEVs are delivered into the venom gland lumen via rough and smooth canals and reach the venom reservoir. Based on immunoelectron microscopy (EM) experiments of the venom protein Lar, we confirmed that IVs are necessary to transport venom proteins in venom gland secretory cells (Fig. 5). Next, we examined the transmission EM of the cross-section of the rough canal and the surrounding microvilli in the venom gland. We found that the *PTCDS*-knockdown *L. heterotoma* had fewer IVs filled with venom proteins than did the control *L. heterotoma* (*P* = 8.9 × 10$^{-5}$, *n* = 5; two-tailed Student's *t* test; Fig. 5; Dataset EV14). Since the MSEVs within the canals are pumped into the venom gland lumen, we further investigated the transmission EM of the section of the venom gland lumen. As expected, we found that the *PTCDS*-knockdown *L. heterotoma* had substantially lower numbers of the MSEVs in the venom gland lumen than the control *L. heterotoma* (*P* = 1.4 × 10$^{-7}$, *n* = 9; two-tailed Student's *t* test; Fig. 5; Dataset EV15). We

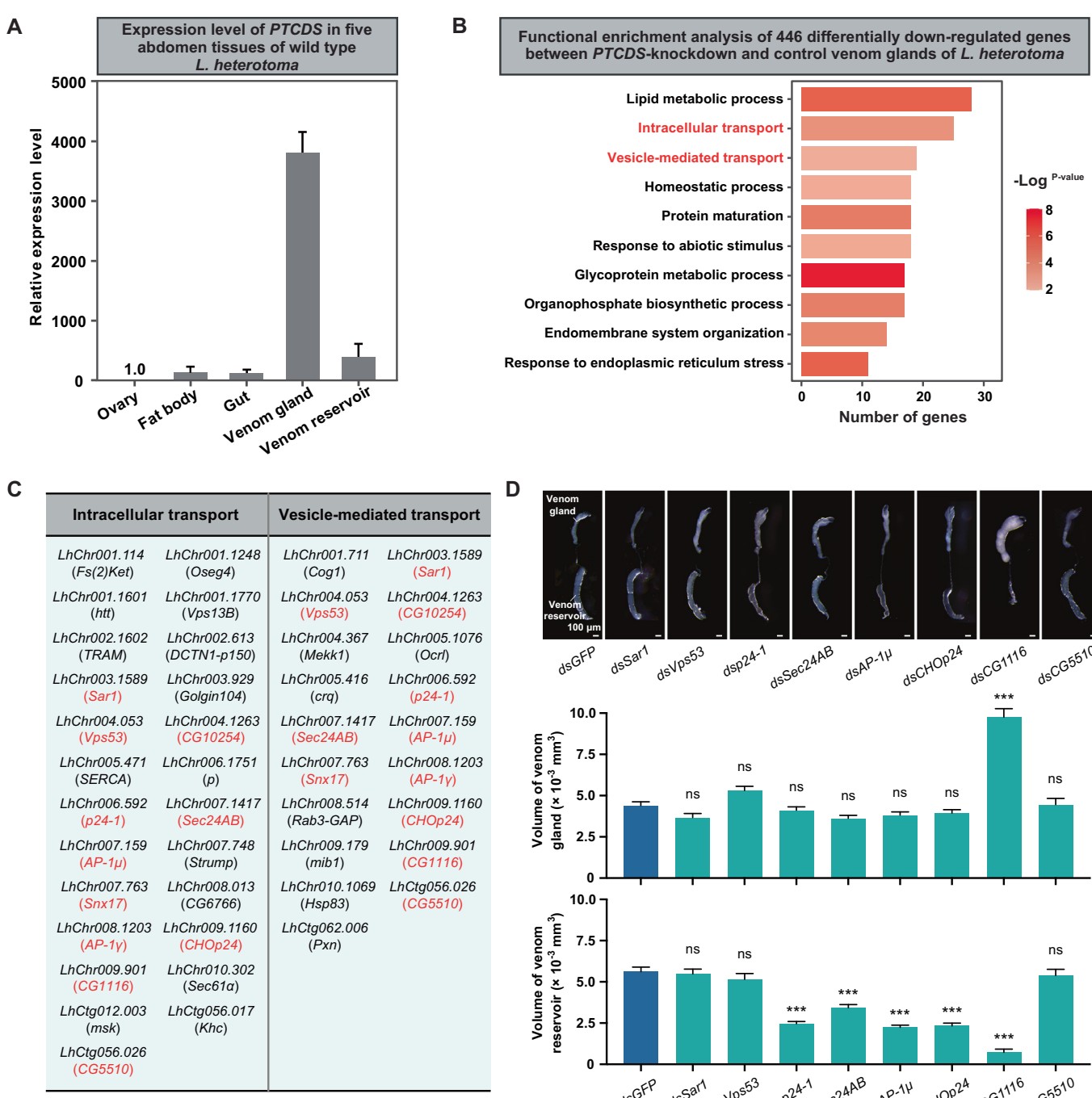

**Figure 4. PTCDS regulates venom secretion into the venom reservoir by modulating protein transport-related genes.**

(A) Relative expression levels of *PTCDS* in five abdomen tissues, including the ovary, fat body, gut, venom gland, and venom reservoir, from 3-day-old wild-type female adult *L. heterotoma*. (B) Functional enrichment analysis of 446 differentially downregulated genes between the *PTCDS*-knockdown and control venom glands of 3-day-old female adult *L. heterotoma*. Statistical significance was assessed using the hypergeometric test, $P < 0.01$. (C) Gene lists involved in "Intracellular transport" (25 genes) and "Vesicle-mediated transport" (19 genes). Red font indicates 11 shared genes between the two groups. (D) Effects of eight protein transport-related genes on venom gland size and venom reservoir size in the knockdown and control parasitoid wasps, respectively. Three-day-old female adult *L. heterotoma* ($n = 30$) were used to examine their venom gland sizes and venom reservoir sizes, respectively. *dsGFP* was used as a control. Data are presented as mean ± SEM. Significance was determined by the Kruskal–Wallis test with Dunn's multiple comparisons test (for venom glands: *dsSar1*: $P = 0.35$; *dsVps53*: $P = 0.21$; *dsp24-1*: $P = 0.99$; *dsSec24AB*: $P = 0.32$; *dsAP-1μ*: $P = 0.91$; *dsCHOp24*: $P = 0.99$; *dsCG1116*: $P = 3.0 \times 10^{-7}$; *dsCG5510*: $P = 0.99$; for venom reservoirs: *dsSar1*: $P = 0.99$; *dsVps53*: $P = 0.99$; *dsp24-1*: $P = 2.0 \times 10^{-9}$; *dsSec24AB*: $P = 3.7 \times 10^{-4}$; *dsAP-1μ*: $P = 6.0 \times 10^{-11}$; *dsCHOp24*: $P = 1.9 \times 10^{-10}$; *dsCG1116*: $P = 1.0 \times 10^{-15}$; *dsCG5510*: $P = 0.99$; ns, not significant; ***$P < 0.001$). Source data are available online for this figure.

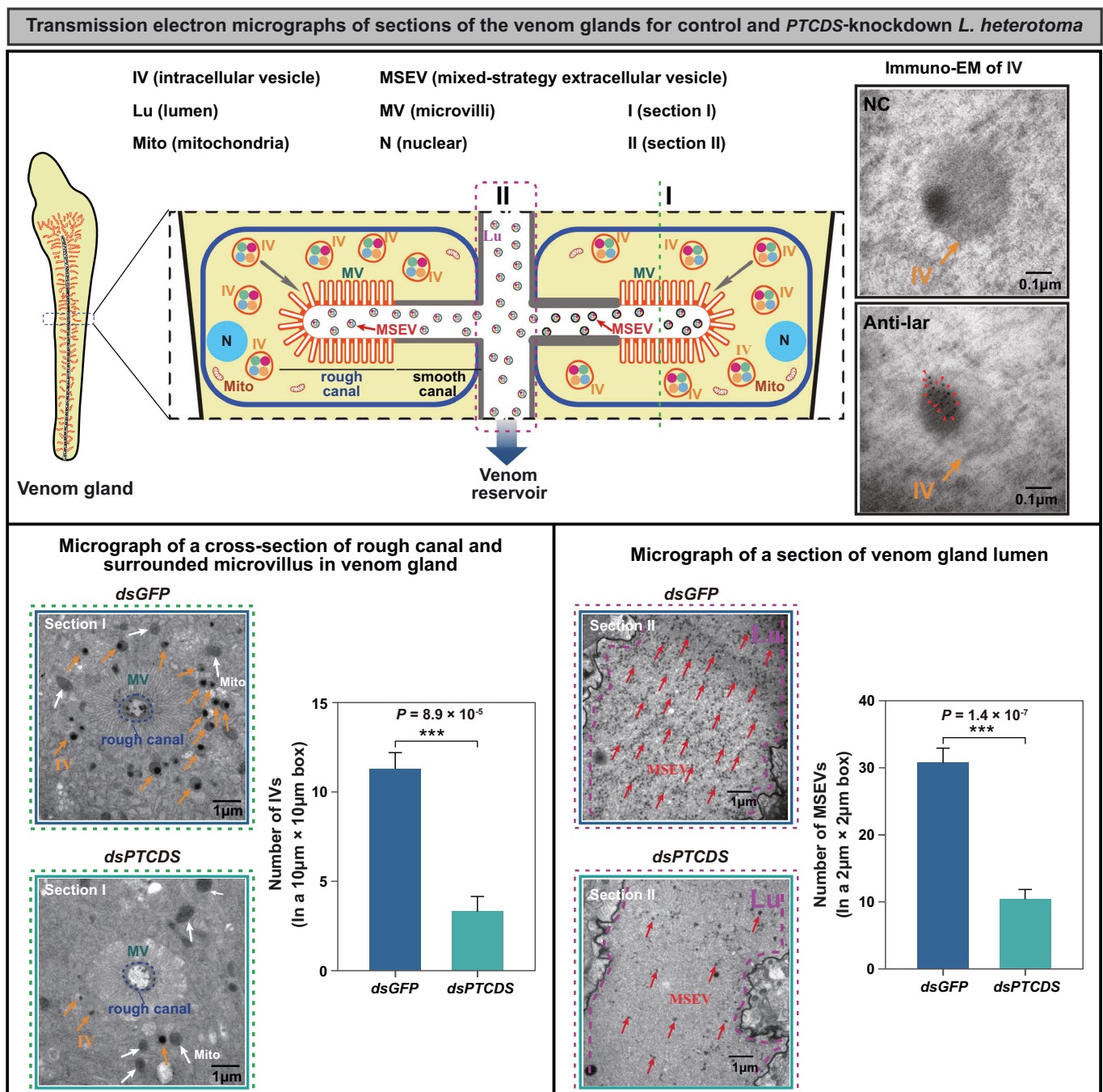

**Transmission electron micrographs of sections of the venom glands for control and *PTCDS*-knockdown *L. heterotoma***

IV (intracellular vesicle)     MSEV (mixed-strategy extracellular vesicle)     Immuno-EM of IV

Lu (lumen)     MV (microvilli)     I (section I)

Mito (mitochondria)     N (nuclear)     II (section II)

Venom gland

Venom reservoir

**Micrograph of a cross-section of rough canal and surrounded microvillus in venom gland**

*dsGFP*

*dsPTCDS*

**Micrograph of a section of venom gland lumen**

*dsGFP*

*dsPTCDS*

further examined the morphology of MSEVs in the venom reservoir, and found that *PTCDS* knockdown in *L. heterotoma* did not alter MSEV morphology. Specifically, the anterior region of the venom reservoir contained aggregated MSEVs consisting 1–6 particles surrounded by a lipid bilayer, while the posterior region retained mature individual MSEVs displaying characteristic 5–6 spikes (Heavner et al, 2017; Rizki and Rizki, 1990) (Fig. EV8). These results revealed that the knockdown of *PTCDS* led to a low number of IVs that transported venom proteins into the canal through the microvillus, which caused a low number of MSEVs, suggesting that the foreign gene *PTCDS* is involved in regulating venom protein secretion in the venom gland of parasitoid wasps.

## The eukaryotic-type CDP-diacylglycerol synthase gene does not regulate stored venom protein amount

The CDP-diacylglycerol synthetase (*CDS*) gene is present in both prokaryotes and eukaryotes, and its core enzymatic function is highly conserved. This enzyme catalyzes the conversion of phosphatidic acid (PA) and cytidine triphosphate (CTP) into cytidine diphosphate-diacylglycerol (CDP-DAG). CDP-DAG then serves as an essential precursor for synthesizing fundamental phospholipids, including phosphatidylglycerol (PG), phosphatidylinositol (PI), and its derivatives (Shen and Dowhan, 1997; Sparrow

◀ **Figure 5. *PTCDS* is responsible for venom protein secretion in venom gland cell.**

Comparison of transmission electron micrographs (TEMs) of different sections (I and II) of the venom glands of the control and *PTCDS*-knockdown 3-day-old female adult *L. heterotoma*. The venom proteins were delivered by intracellular vesicles (IVs) to the microvillar region of the rough canals in venom gland secretory cells, which was confirmed by immuno-EM experiments of the venom protein Lar in this study. Red arrowheads point to the Lar proteins in IVs. The venom proteins are secreted into the canal through microvilli (MVs), where they are reassembled and reloaded into specialized mixed-strategy extracellular vesicles (MSEVs). The left two panels are micrographs of Section I, which represent the canal and the microvillus surrounded by the IVs filled with the venom proteins in the *PTCDS*-knockdown and control venom glands, respectively. Yellow arrowheads point to the IVs filled with the venom proteins. The green text indicates the position of the MVs. The canal is outlined by a blue-dotted circle. The white arrowheads point to the mitochondria (Mito). The right two panels are micrographs of Section II, which represent the venom gland lumen in the *PTCDS*-knockdown and control venom glands. The MSEVs from the venom gland lumen are pumped into the venom reservoir. The lumen is outlined by a purple-dotted line. The red arrowheads point to representative MSEVs in the lumen. The numbers of IVs were measured from areas around the rough canals (10 μm × 10 μm) in the micrographs of the *PTCDS*-knockdown and control wasps. Five different images were examined for each group. The number of MSEVs was measured from each of three randomly selected areas (2 μm × 2 μm) in the micrographs of the *PTCDS*-knockdown and control wasps. Three different images were examined for each group. Data are presented as mean ± SEM. Significance was determined by two-tailed Student's *t* test (\*\*\**P* < 0.001). Source data are available online for this figure.

and Raetz, 1985; Blunsom and Cockcroft, 2020). Consistent with its fundamental role in lipid signaling and metabolism, studies have shown that *CDS* is involved in diverse physiological processes such as the visual phototransduction system (Wu et al, 1995), vascular development (Zhao et al, 2019), and the regulation of cell growth and fat storage (Liu et al, 2014). We next identified the eukaryote-type *CDS* gene in all five parasitoid wasp species (*L. heterotoma, L. syphax, L. drosophilae, L. boulardi,* and *L. myrica*). The encoded eukaryotic-type CDS proteins exhibited high sequence conservation, sharing 97% amino acid identity. Although eukaryotic-type CDS shares only ~27% amino acid identity with the PTCDS in *L. heterotoma* and *L. syphax*, all homologs contain a CDP-diacylglycerol synthetase domain, including a conserved CDS signature motif (KDX$_5$PGHGGX$_2$DRXD) (Fig. EV9).

To investigate the functional role of the eukaryotic-type *CDS*, we performed RNAi-mediated knockdown of the gene in female wasps of *L. heterotoma, L. syphax,* and *L. boulardi*. qRT-PCR analysis confirmed highly efficient silencing, with knockdown efficiencies of 95% ± 1.4% in *L. heterotoma*, 62% ± 2.0% in *L. syphax*, and 96% ± 0.5% in *L. boulardi* (Fig. EV10A). Compared to controls, eukaryote-type *CDS* knockdown wasps showed no significant differences in venom gland size (*L. heterotoma*: *P* = 0.71, *n* = 30; *L. syphax*: *P* = 0.42, *n* = 23; *L. boulardi*: *P* = 0.43, *n* = 30; Mann–Whitney *U* test for *L. heterotoma*, Welch's *t* test for *L. syphax*, two-tailed Student's *t* test for *L. boulardi*) or venom reservoir size (*L. heterotoma*: *P* = 0.55, *n* = 30; *L. syphax*: *P* = 0.60, *n* = 23; *L. boulardi*: *P* = 0.46, *n* = 30; two-tailed Student's *t* test) (Fig. EV10B; Dataset EV16).

These results show that the eukaryote-type *CDS* is not responsible for venom protein secretion in the venom gland of these parasitoid wasps. This further suggests that the HGT-acquired *PTCDS* gene in *L. heterotoma* and *L. syphax* may have been co-opted for a novel, venom gland-specific function, potentially through a mechanism independent of its canonical role in phospholipid biosynthesis.

## Discussion

Genome sequencing data have recently demonstrated that eukaryotic genome evolution has been remarkably influenced by the occurrence of horizontal gene transfer (HGT) (Dunning Hotopp, 2011; Keeling and Palmer, 2008; Ochman et al, 2000; Li et al, 2022; Shen et al, 2018). In this study, we systematically examined putative HGT-acquired genes in five species of *Leptopilina* wasps and identified a total of 19 foreign genes that were horizontally acquired via five distinct transfer events from bacteria (Fig. 1A,B). To date,

Li et al carried out the most comprehensive investigation of HGTs in 218 insect genomes and reported 182 HGT-acquired genes in 68 hymenopterans (Li et al, 2022), but they did not include *Leptopilina* wasps (the date of collection of the 218 insect genomes was earlier than the release date of the *Leptopilina* genomes). Interestingly, when comparing the list of 182 previously published HGT-acquired genes in 68 hymenopterans, we found that four of the 19 foreign genes that involve HGT events3-5 were specifically acquired by the genus *Leptopilina*, while five HGT-acquired genes that involve HGT event1 were acquired by the last common ancestor of all 68 hymenopterans (Fig. 1).

Parasitoid–host interactions can constitute an arms race between successful parasitism and host resistance to parasitism. Parasitoid wasps use venom proteins to increase their reproductive success in or on hosts, while hosts mobilize their immune defence to suppress parasitoid wasp attacks (Asgari and Rivers, 2011; Moreau and Asgari, 2015; Poirié et al, 2014). In addition to the host immune defence strategy, a handful of studies have experimentally shown that hosts capture foreign genes from bacteria and viruses via HGT to increase their defence against parasitoid wasps' attacks (Di Lelio et al, 2019; Gasmi et al, 2021; Verster et al, 2023). Two recent and well-studied papers have shown that HGTs are important for resistance to wasp parasitism; a parasitoid killing factor gene transferred from a virus to lepidopterans contributes to the killing capacity against parasitoid wasp attacks (Gasmi et al, 2021), whereas two toxic genes transferred from bacteria to fruit flies contribute to toxic defence against parasitoid wasp attacks (Verster et al, 2023).

What has yet to be tested is whether parasitoid wasps receive and benefit from HGT-acquired genes. In our study, we identified a foreign gene, the prokaryote-type CDP-diacylglycerol synthase (*PTCDS*), which was transmitted into the last common ancestor of the parasitoid wasps *L. heterotoma* and *L. syphax* from the bacterial family Rickettsiaceae. We found that in *L. heterotoma* and *L. syphax*, the HGT-acquired gene *PTCDS* knockdown led to a lower efficiency of the secretion of venom proteins from venom gland secretory cells into the venom gland lumen, which in turn reduced the amount of stored venom and the size of the venom reservoir. While our data suggest a model in which *PTCDS* functions within the venom gland cells to regulate vesicle-mediated secretion, we cannot rule out the alternative possibility that *PTCDS* regulates venom reservoir size, which in turn could affect the amount of stored venom protein, possibly through yet-unknown mechanisms that influence the venom gland's activity. Furthermore, as the stored venom protein in venom reservoir is dynamically

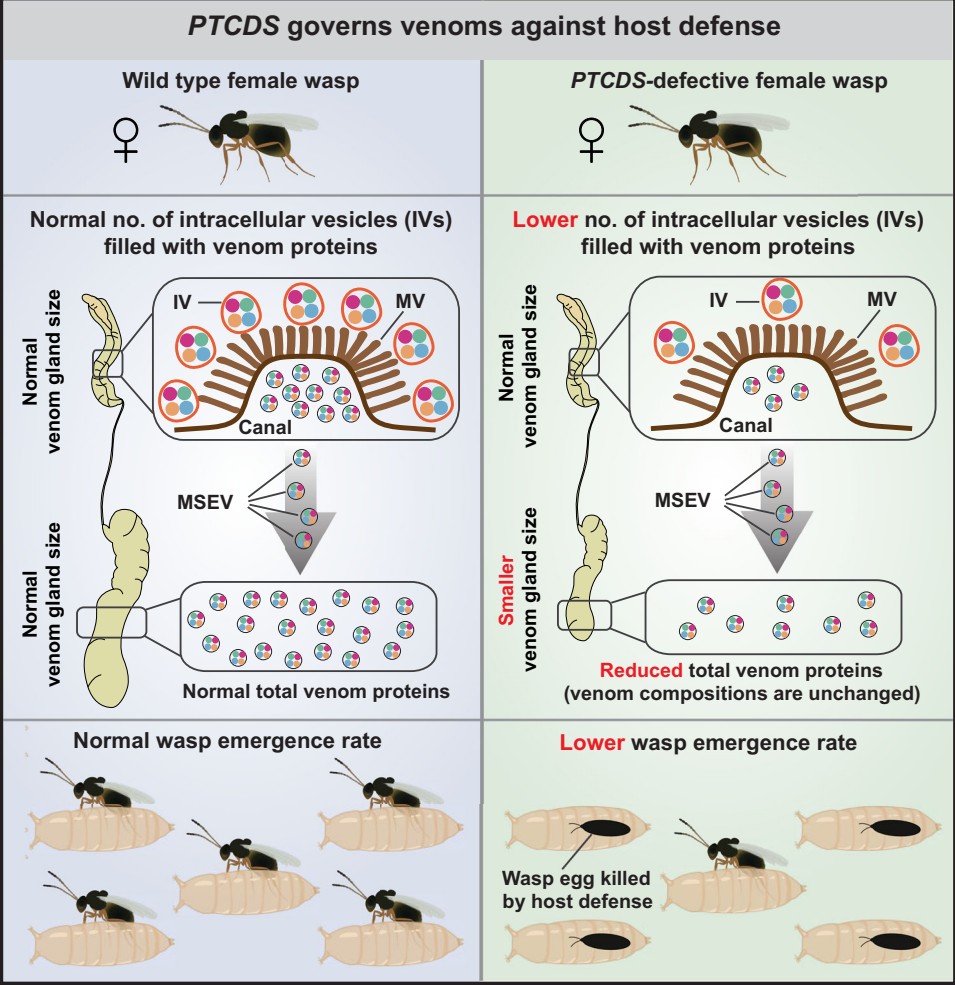

**Figure 6.   A schematic model of the role of HGT-acquired *PTCDS* in parasitoid wasps.**

A proposed model of the role of the HGT-acquired gene *PTCDS* in female parasitoid wasps. Dysfunction of the HGT-acquired gene *PTCDS* leads to a lower number of intracellular vesicles (IVs) filled with venom proteins that are secreted into the canal through microvilli (MVs) in venom gland cells, which in turn contributes to a reduced amount of total venom proteins (but does not change their composition) in the venom reservoir. Consequently, *PTCDS*-defective female parasitoid wasps, which are equipped with a reduced amount of total venom proteins, have a substantially lower emergence rate of wasp offspring because of their lower ability to suppress host cellular immune defence.

replenished after use, the secretion state of the gland is not static. Therefore, future work is needed to elucidate the precise role of *PTCDS* in these dynamic processes, specifically whether it contributes to the secretory machinery, modulates venom reservoir development, or both. These observations also raise important questions about venom regulation in *PTCDS*-deficient *Leptopilina* species, including *L. drosophilae*, *L. boulardi*, and *L. myrica*. Since these species lack the *PTCDS* gene that mediates venom secretion in *L. heterotoma* and *L. syphax*, they must employ alternative mechanisms to regulate venom production and storage. Two non-mutually exclusive hypotheses could explain this phenomenon: first, these species may have evolved to require significantly lower venom quantities for successful parasitism; second, they may

possess distinct, *PTCDS*-independent molecular pathways that control venom gland secretion and reservoir storage. Further comparative studies of venom regulation mechanisms across *Leptopilina* species will be crucial to elucidate these evolutionary adaptations. Nevertheless, all the experimental evidence shows that the foreign gene *PTCDS* governs venom proteins against host defence, ensuring the successful growth and development of offspring within the host larvae (Fig. 6). Whereas previously well-studied HGT-acquired genes in the arms race between parasitoid wasps and hosts occurred in hosts, increasing host fitness (Di Lelio et al, 2019; Gasmi et al, 2021; Verster et al, 2023), our results suggest that parasitoid wasps recruited a foreign gene as a reaction to increase their fitness.

# Methods

### Reagents and tools table

| Reagent/resource | Reference or source | Identifier or catalog number |
|---|---|---|
| **Experimental models** | | |
| *L. heterotoma* | Chen et al, 2021 | N/A |
| *L. syphax* | Dong et al, 2025b | N/A |
| *L. drosophilae* | Zhang et al, 2023 | N/A |
| *L. boulardi* | Chen et al, 2021 | N/A |
| *L. myrica* | Dong et al, 2025a | N/A |
| *D. melanogaster (w¹¹¹⁸)* | Bloomington *Drosophila* Stock Center | Cat#BL5905 |
| *D. melanogaster (Hml-GAL4 > UAS-GFP)* | Bloomington *Drosophila* Stock Center | Cat#BL30140 |
| *D. simulans* | Core Facility of *Drosophila* Resource and Technology, Chinese Academy of Sciences | Cat#BCF93 |
| *D. sechellia* | KYORIN-Fly, Fly Stocks of Kyorin University | Cat#K-S10 |
| *D. pseudoobscura* | Core Facility of *Drosophila* Resource and Technology, Chinese Academy of Sciences | Cat#BCF95 |
| *D. mauritiana* | Chen et al, 2021 | N/A |
| *D. santomea* | Pang et al, 2024 | N/A |
| **Recombinant DNA** | | |
| N/A | N/A | N/A |
| **Antibodies** | | |
| Anti-Lar primary antibody | Huang et al, 2021 | N/A |
| HRP-conjugated Goat anti-Rabbit IgG (H + L) | ABclonal | AS014 |
| Anti-Rabbit IgG (whole molecule) –Gold antibody produced in goat | Sigma | G7402 |
| **Oligonucleotides and other sequence-based reagents** | | |
| PCR primers | This study | Dataset EV17 |
| **Chemicals, enzymes and other reagents** | | |
| FastPure Cell/Tissue Total RNA Isolation Kit | Vazyme | RC-101 |
| HiScript III RT SuperMix for qPCR Kit | Vazyme | R223-01 |
| 2 × Phanta Max Master Mix | Vazyme | P515-02 |
| MiniBEST Agarose Gel DNA Extraction Kit Ver.4.0 | Takara | Cat#9762 |
| T7 High Yield RNA Transcription Kit | Vazyme | TR101-02 |
| Minute™ Total Protein Extraction Kit for Animal Cultured Cells/Tissues | Invent Biotechnologies | SD-001/SN-002 |
| Coomassie Brilliant Blue R-250 | Solarbio | Cat#C8430 |
| Pierce™ BCA protein assay kit | Thermo Scientific | REF23225 |
| ChamQ SYBR qPCR Master Mix Kit | Vazyme | Q311-02 |
| 20X PBS Buffer | Sangon Biotech | Cat# B548117-0500 |
| Agarose | Tsingke | Cat# TSJ001 |
| Bovine serum albumin | Sigma | Cat# V900933-100G |
| Phosphate buffer | Sinopharm Chemical Reagent Co., Ltd. | C258590010 |
| Glutaraldehyde | Sinopharm Chemical Reagent Co., Ltd. | 30092436 |
| Osmic acid | SPI-CHEM | N/A |
| Ethanol | Sinopharm Chemical Reagent Co., Ltd. | 10009259 |
| Acetone | Sinopharm Chemical Reagent Co., Ltd. | 10000418 |
| Spurr resin | SPI-CHEM | N/A |
| Uranyl acetate | SPI-CHEM | N/A |
| Alkaline lead citrate | Sinopharm Chemical Reagent Co., Ltd. | N/A |
| **Software** | | |
| GraphPad v8.0 | https://www.graphpad.com/features | |
| MASCOT v2.2 | Perkins et al, 1999 | |
| Trimmomatic v0.39 | Bolger et al, 2014 | |
| featureCounts v2.0.1 | Liao et al, 2014 | |
| R packages DESeq2 v1.30.1 | Liao et al, 2014 | |
| STAR v2.7.10a | Dobin et al, 2013 | |
| Metascape v3.5 | Zhou et al, 2019 | |
| BUSCO, v5.2.2 | busco.ezlab.org | |
| HGTfinder v1 | Li et al, 2022 | |
| Salmon v0.12.0 | Patro et al, 2017 | |
| ImageJ | http://imagej.nih.gov/ij/ | |
| **Other** | | |
| C18 cartridges | Empore SPE Cartridges C18 | |
| C18 reversed-phase analytical column | Thermo Fisher Scientific Easy Column | |

## Taxon sampling

To systematically identify the putative HGT-acquired genes in the *Leptopilina* parasitoid wasps that attack *Drosophila* hosts, we downloaded five publicly available genome assemblies (*L. heterotoma*, *L. syphax*, *L. drosophilae*, *L. boulardi*, and *L. myrica*) and their annotations from GenBank data (Dataset EV1). We used Benchmarking Universal Single-Copy Orthologs (BUSCO, v5.2.2), to assess the quality of each of the five parasitoid wasp genomes and

found that the completeness of all the assemblies was greater than or equal to 97% of the 1367 full-length BUSCO nuclear genes in Insecta. On the basis of five annotation files, we extracted the amino acid sequences of all 74,156 protein-coding genes for identifying putative HGT-acquired genes. Detailed information on the genome information and characteristics of each species is given in Dataset EV1.

## Identification of HGTs and their expression profiles

### Identification of HGTs

We used the HGTfinder v1, which is a robust and conservative phylogeny-based approach (Li et al, 2022), to assess whether each of 74,156 protein-coding genes had been horizontally acquired from non-metazoan organisms. In brief, this approach incorporated the information from each gene's Alien Index (AI) score, which compared the similarity of the gene between specified ingroup and outgroup taxa (e.g., insects and bacteria, respectively), the distribution of outgroup taxa in the list of each gene's top 1000 blast hits against the Refseq database (last accessed May 3, 2022), as well as each gene's placement in a maximum likelihood phylogenic tree with its 1000 most similar homologs. From the 74,156 genes analyzed, we identified 19 putative HGT-acquired genes from the bacterial organisms. The maximum likelihood phylogenetic trees for all 19 identified HGT candidates and the sequence alignments have been deposited in the Figshare repository (https://figshare.com/s/801551d66a83eaaafde9) and are publicly available. Given its central role in this study, the phylogenetic tree for the *PTCDS* gene is provided as Fig. EV1.

### Expression profiles of HGTs

We generated new transcriptome data from seven different tissues to investigate the expression profiles of the 19 HGT-acquired genes. Tissues were dissected from the following numbers of individuals to ensure sufficient RNA yield: antennae (from 100 wasps), heads (from 50 wasps), thoraxes (from 40 wasps), abdomens (from 40 wasps), legs (from 270 wasps), ovaries (from 50 wasps), and venom glands (from 100 wasps). For a given tissue, total RNA was isolated from 3-day-old *L. heterotoma*, *L. syphax*, *L. drosophilae*, *L. boulardi*, and *L. myrica* via the FastPure Cell/Tissue Total RNA Isolation Kit (Vazyme). For the RNA-seq data, library construction and sequencing were performed on an Illumina HiSeq2000 platform (with paired ends). Low-quality reads and adapter sequences were removed from the raw RNA-seq reads via Trimmomatic v0.39 (Bolger et al, 2014) with the default parameters. Clean reads were mapped to the reference genome via STAR v2.7.10a (Dobin et al, 2013). The number of reads mapped to each gene was determined via featureCounts v2.0.1 (Liao et al, 2014). Expression profiles across different tissues and different parasitoid wasps were determined as normalized transcripts per million (TPM) using salmon v0.12.0 (Patro et al, 2017). To compare HGT-acquired gene expression levels across different tissues, TPM values were normalized using a two-step approach: (i) tissue-specific max-scaling, where values were divided by the maximum expression level with each tissue sample, followed by (ii) cross-tissue Z-score standardization of the scaled values. HGT-acquired genes and their relative expression levels are provided in Dataset EV2. Although this transcriptome analysis did not include sequenced biological replicates, the high-level expression of *PTCDS*

in the venom gland was validated by subsequent qRT-PCR experiments performed with three independent biological replicates.

## Knockdown of the HGT-acquired gene *PTCDS* in parasitoid wasps

### Animal rearing

The parasitoid wasps *L. boulardi* and *L. heterotoma* were kindly provided by Dr. Dan Hultmark and Dr. István Andó (Chen et al, 2021). The remaining three parasitoid wasps *L. syphax* (COI: GenBank accession number OM272847), *L. drosophilae* (COI: GenBank accession number OM328085), and *L. myrica* (COI: GenBank accession number OP013292.1) were collected from traps at Taizhou, Zhejiang Provence, China (Zhang et al, 2023; Dong et al, 2025a; Dong et al, 2025b). All parasitoid wasps were maintained on the *Drosophila melanogaster* (*w*^*1118* strain) host, and the adult wasps were provided with apple juice/agar medium (27 g agar, 33 g brown sugar and 330 mL pure apple juice in 1000 ml diluted water) for further experiments.

### RNAi assay

For RNAi-mediated knockdown of the *PTCDS* gene, the coding regions of *LhChr005.253* (*LhPTCDS*) in *L. heterotoma* and *LsChr004.1071* (*LsPTCDS*) in *L. syphax* were amplified with the forward primers (*LhPTCDS*: 5'-GGTTGAAGCCTCTTTTCCACC-3'; *LsPTCDS*: 5'-TGGTTGAAGCCTCTTTTCTACCT-3') and the reverse primers (*LhPTCDS*: 5'-CACTCCTCCATGACCCGGTA-3'; *LsPTCDS*: 5'-CCAACACTCCTCCATGACCC-3') from the cDNA with 2 × Phanta Max Master Mix (Vazyme), respectively. The *GFP* gene was used as the control. The dsRNA templates were amplified with T7 promoter sequence primers. The primers used in this experiment are listed in Dataset EV17. Double-stranded RNA was synthesized via a T7 High Yield RNA Transcription Kit (Vazyme) according to the manufacturer's protocol. Approximately 20 nl of dsRNA (5 μg/μL) was injected into the abdomen of each fifth-instar wasp larva via an Eppendorf FemtoJet 4i Microinjector. Three biological replicates were checked for the efficiency of RNAi. The housekeeping gene *tubulin* was used as an internal control for data normalization. The data were analysed via the $2^{-\triangle\triangle CT}$ method. After the dsRNA-treated parasitoid wasps emerged from the hosts, we used adult females at the normal age of maturity, which were 3-day-old female adult *L. heterotoma* and 9-day-old female adult *L. syphax*, to conduct the experiments described below. The same RNAi protocol was conducted for those protein transport-related genes, with all primer sequences documented in Dataset EV17.

### Growth and development

To investigate the growth and development of the *PTCDS*-knockdown and control *L. heterotoma* and *L. syphax* wasps, 3-day-old female adult *L. heterotoma* ($n = 30$) and 9-day-old female adult *L. syphax* ($n = 25$) were dissected in 1× phosphate-buffered saline (PBS) and photographed with a digital microscope SZX2-ILLT (OLYMPUS). Body size and ovary size were measured via ImageJ software (http://imagej.nih.gov/ij/).

### Host-searching behaviours

Y-tube behavioural assays and host-searching time assays (Sheng et al, 2023) were used to evaluate the host-searching behaviours of

the dsRNA-treated *L. heterotoma* and *L. syphax* female adults. For the Y-tube behavioural assay, a group of 16 3-day-old *L. heterotoma* female adults or 9-day-old *L. syphax* female adults were placed in the bottom of the central arm of a Y-tube. The food containing the host larvae was put into one of the choice arms of the Y-tube, and 2% agarose gel was put into another arm as a control. The inlet air was then pushed into each choice arm of the Y-tube at a rate of 100 ml per min. We recorded the number of female adults in each choice arm after 10 min. Six biological replicates were performed for each treatment. For the host-searching time assay, a group of 20 *Drosophila melanogaster* host larvae were placed onto a 35 mm dish with standard cornmeal/molasses/agar medium. A 3-day-old *L. heterotoma* or 9-day-old *L. syphax* female adult was subsequently released into the dish. The time at which the wasps found the host larva was recorded. Fifteen biological replicates were performed for each treatment.

### Oviposition assay

The 3-day-old *PTCDS*-knockdown or control *L. heterotoma* female adults were allowed to parasitize second-instar *Drosophila* larvae at a wasp/host ratio of 3:360 for 12 h. The 9-day-old *PTCDS*-knockdown or control *L. syphax* female adults were allowed to parasitize first-instar *Drosophila* larvae at a wasp/host ratio of 3:180 for 6 h. After that, the female wasps were removed, and the host larvae were dissected to detect whether they had been parasitized. Oviposition rate = (the number of host larvae parasitized by the wasps/the number of total host larvae). Six different *Drosophila* species (*D. melanogaster*, *D. simulans*, *D. sechellia*, *D. pseudoobscura*, *D. mauritiana*, and *D. santomea*) were used as the hosts. Three biological replicates were performed for each treatment.

### Wasp emergence assay

The 3-day-old *PTCDS*-knockdown or control *L. heterotoma* female adults were allowed to parasitize second-instar *Drosophila* larvae at a wasp/host ratio of 3:360 for 12 h. The 9-day-old *PTCDS*-knockdown or control *L. syphax* female adults were allowed to parasitize first-instar *Drosophila* larvae at a wasp/host ratio of 3:180 for 6 h. After that, the female wasps were removed, and the host larvae were maintained at 25 °C to calculate the emergence rate of the wasp offspring. The emergence rate of wasp offspring = the number of wasp offspring that successfully emerged from the host larvae/the number of total host larvae. Six different *Drosophila* species (*D. melanogaster*, *D. simulans*, *D. sechellia*, *D. pseudoobscura*, *D. mauritiana*, and *D. santomea*) were used as hosts. Three biological replicates were performed for each treatment.

## Cellular immunity of *D. melanogaster* host larvae in response to parasitoid wasps

### Encapsulation response

To examine the lower emergence rates of *PTCDS*-knockdown wasps, we investigated the encapsulation rates of *D. melanogaster* larvae parasitized by *PTCDS*-knockdown wasps and control wasps. The *D. melanogaster* larvae were parasitized via the same conditions described above for the wasp emergence assay. The female wasps were removed, and the host larvae were maintained at 25 °C to examine the encapsulation response. The number of fly host larvae containing encapsulated wasp eggs was recorded. Encapsulation rate = (the number of host larvae with encapsulated

wasp egg/the number of total host larvae). Three biological replicates were performed for each treatment.

### Haemocyte measurement

To quantify the number of haemocytes in the host larvae parasitized by the *PTCDS*-knockdown wasps and control wasps, we used *Hml-GAL4 > UAS-GFP* (BDSC stock #30140) *D. melanogaster* as the host. This haemocyte-specific *GAL4* and fluorescent reporter fusion construct labelled all circulating haemocytes (e.g., plasmatocytes, crystal cells, and lamellocytes). The parasitized host larvae (96 h after egg hatching) were carefully rinsed three times with 1×PBS and dried with filter paper before dissection. Then, the haemolymph of ten host larvae was diluted in 20 µL of 1×PBS, and 8 µL of the mixture was dropped on a hemocytometer (Watson). The number of *GFP*-labelled haemocytes was counted under a Zeiss LSM 800 confocal microscope. Ten biological replicates were performed for each treatment.

## Venom apparatus size measurement

The 3-day-old *PTCDS*-knockdown or control *L. heterotoma* female adults and 9-day-old *PTCDS*-knockdown or control *L. syphax* female adults were dissected in 1×PBS, and the venom systems (venom gland and venom reservoir) were photographed with a digital microscope SZX2-ILLT (OLYMPUS). For the venom gland size and the venom reservoir size, the diameter at three different positions (near the ovipositor, in the middle, and away from the ovipositor) was measured via ImageJ software (http://imagej.nih.gov/ij/). The length (L) and average radius (r) of the three measurements along the venom gland and reservoir were used to obtain the volume (V) via the following formulas (Lemauf et al, 2021): $V = \pi r^2 L$. Thirty and 25 biological replicates were performed for *L. heterotoma* and *L. syphax* for each treatment, respectively.

## Characteristics of venom proteins from the venom reservoir of the parasitoid wasp *L. heterotoma*

We investigated the amount of total venom proteins from the venom reservoir between the *PTCDS*-knockdown and control female adult wasps via SDS-PAGE, quantitative protein assay kit, and mass spectrometry (LC-MS/MS).

### SDS-PAGE

Twenty venom reservoirs from 3-day-old *PTCDS*-knockdown or control *L. heterotoma* female wasps were pierced in a 20 µl drop of 1 × PBS to collect venom proteins. Total venom proteins were subsequently extracted and purified via the Minute™ Total Protein Extraction Kit for Animal Cultured Cells/Tissues (Invent Biotechnologies) according to the manufacturer's protocol. Total venom proteins from 20 venom reservoirs were visualized via SDS–polyacrylamide gel electrophoresis analysis with Coomassie Brilliant Blue R-250 staining (Solarbio).

### Quantitative total venom proteins

A single venom reservoir from a 3-day-old *PTCDS*-knockdown or control *L. heterotoma* female wasp was pierced in a 20 µl drop of 1×PBS to collect venom proteins. Total venom proteins per venom reservoir were subsequently extracted and purified via the Minute™ Total Protein Extraction Kit for Animal Cultured Cells/Tissues

(Invent Biotechnologies) according to the manufacturer's protocol. The amount of total venom proteins per venom reservoir was measured via a Pierce™ BCA protein assay kit (Thermo Scientific) according to the manufacturer's protocol. Thirty biological replicates were performed for each treatment.

### Identification of venom proteins via LC-MS/MS

The 200 venom reservoirs from 3-day-old *PTCDS*-knockdown or control *L. heterotoma* female wasps were dissected in 1×PBS on an ice plate and washed three times in 1×PBS. The venom reservoirs were then pierced in a cell culture dish, and the venom proteins were collected in an Eppendorf tube. After centrifugation at $3000 \times g$ at 4 °C for 1 min, the supernatant (pure venom proteins) was used for liquid chromatography–tandem mass spectrometry (LC-MS/MS) experiments. The venom proteins were dissolved in 100 µl SDT lysis buffer (4% SDS, 100 mM Tris-HCl, and 1 mM DTT, pH 7.6). The sample was then boiled for 15 min and centrifuged at $3000 \times g$ at 4 °C for 40 min. The supernatant was collected into a new Eppendorf tube. The detergent and DTT were removed by repeated ultrafiltration (Microcon units) using UA buffer (8 M urea, 150 mM Tris-HCl, pH 8.0), followed by incubation with 100 µl iodoacetamide (100 mM) for 30 min in the dark to block reduced cysteine residues. Then, the protein suspensions were digested with 3 µg trypsin (Promega) in 40 µl 100 mM $NH_4HCO_3$ buffer overnight at 37 °C. The resulting peptides were desalted on C18 cartridges (Empore SPE Cartridges C18 (standard density), bed I.D. 7 mm, Volume 3 ml, Sigma), concentrated by vacuum centrifugation, and reconstituted in 40 µl of 0.1% (v/v) formic acid. LC-MS/MS analysis was performed on a Q Exactive mass spectrometer (Thermo Fisher Scientific) coupled to an Easy nLC (Thermo Fisher Scientific). A 6-µl aliquot of the peptide mixture was loaded onto a reverse-phase trap column (Thermo Fisher Scientific Acclaim PepMap100, 100 µm × 2 cm, nanoViper C18) connected to a C18 reversed-phase analytical column (Thermo Fisher Scientific Easy Column, 10 cm long, 75 µm inner diameter, 3 µm resin) in buffer A (0.1% formic acid) and separated with a linear gradient of buffer B (84% acetonitrile and 0.1% formic acid) at a flow rate of 300 nl/min. The eluted peptides were ionized, and the full MS spectrum (from *m/z* 300 to 1800) was acquired via a precursor ion scan using the Orbitrap analyzer with a resolution of $r = 70,000$ at *m/z* 200, followed by 20 MS/MS events in Orbitrap analysis with a resolution of $r = 17,500$ at *m/z* 200. The raw MS data were searched against a protein database derived from the *L. heterotoma* genome assembly GCA_032872495.1 using MASCOT v2.2 (Perkins et al, 1999). MS/MS tolerance was set at 20 ppm. Trypsin was specified as the proteolytic enzyme, allowing for up to two missed cleavages. Carbamidomethylation of cysteine was set as a fixed modification, while oxidation of methionine and acetylation of protein N-termini were included as variable modifications. False discovery rate (FDR) of ≤1% was applied for protein identification. Venom proteins were identified using an integrated transcriptomic and proteomic approach, as previously described (Huang et al, 2021; Martinson et al, 2017). A protein was defined as a venom component if it met the following criteria: (i) it was supported by at least three aligned proteomic peptides, and (ii) its corresponding gene ranked among the top 500 most highly expressed genes in the venom gland transcriptome.

### Western blot analysis of the venom protein Lar

Twenty venom glands and 20 venom reservoirs from 3-day-old *PTCDS*-knockdown or control *L. heterotoma* female wasps were used to collect the total venom proteins via the Minute™ Total Protein Extraction Kit for Animal Cultured Cells/Tissues (Invent Biotechnologies). Here, we specifically examined the well-known venom protein Lar (lymph gland apoptosis-related protein) reported in a previous study (Huang et al, 2021), which was used as a venom protein marker. Venom proteins from ten venom glands and one venom reservoir were reserved for SDS–polyacrylamide gel electrophoresis analysis, respectively. The proteins were subsequently transferred to a polyvinylidene difluoride membrane (Millipore). The membrane was incubated at 4 °C overnight with a primary Lar-specific antibody (1:1000). The horseradish peroxidase-conjugated anti-rabbit IgG secondary antibody (ABclonal) was used at a dilution of 1:2000 to display the venom protein marker Lar.

## Quantitative real-time PCR

Five different tissues (gut, ovary, fat body, venom gland and venom reservoir) from 3-day-old wild-type *L. heterotoma* female adults were sampled. Total RNA was extracted via the FastPure Cell/Tissue Total RNA Isolation Kit (Vazyme) and then reverse transcribed into cDNA via the HiScript III RT SuperMix for qPCR Kit (Vazyme) according to the manufacturer's protocol. Quantitative real-time PCR (qRT-PCR) was performed with the QuantStudio3 Real-Time PCR System (Thermo Fisher Scientific) with the ChamQ SYBR qPCR Master Mix Kit (Vazyme). Reactions were carried out for 30 s at 95 °C, followed by 45 cycles of three-step PCR for 10 s at 95 °C, 20 s at 55 °C, and 20 s at 72 °C. The RNA levels of the target genes were normalized to that of *tubulin* mRNA, and the relative concentration was determined via the $2^{-\Delta\Delta CT}$ method. Three biological replicates were performed for each tissue. The primers used in this experiment are listed in Dataset EV17.

## Transcriptome data

### RNA sequencing

We collected 400 venom glands from 3-day-old *PTCDS*-knockdown and control *L. heterotoma* adult females to generate transcriptome data. Total RNA was isolated via the FastPure Cell/Tissue Total RNA Isolation Kit (Vazyme) according to the manufacturer's protocol. For the RNA-seq data, library construction and sequencing were performed on an Illumina HiSeq2000 platform (pair ends).

### Transcriptome analysis

Raw RNA-seq reads were removed from low-quality reads and adapter sequences via Trimmomatic v0.39 (Bolger et al, 2014) with default parameters. Clean reads were mapped to the reference genome via STAR v2.7.10a (Dobin et al, 2013). The number of reads mapped to each gene was determined via featureCounts v2.0.1 (Liao et al, 2014). The resulting transcript counts were subjected to R packages DESeq2 v1.30.1 (Liao et al, 2014). Differentially expressed genes with a *P* value of ≤0.05 and a log2-fold change of ≥1 or ≤ −1 were subjected to functional enrichment analysis via Metascape v3.5 (https://metascape.org/) (Zhou et al, 2019).

## Electron microscopy of the venom glands of the parasitoid wasp *L. heterotoma*

Approximately 30 venom glands from 3-day-old *PTCDS*-knockdown and control *L. heterotoma* female wasps were dissected and fixed with 2.5% glutaraldehyde in phosphate buffer (0.1 M, pH 7.0) overnight. The samples were subsequently washed three times in phosphate buffer (0.1 M, pH 7.0) for 15 min at each step. After being postfixed with 1% osmic acid in phosphate buffer for 2 h, the samples were washed three times in phosphate buffer (0.1 M, pH 7.0) for 15 min at each step. The samples were dehydrated with a graded series of ethanol (30%, 50%, 70%, 80%, 90% and 95%) for ~15 min at each step and then dehydrated with alcohol for 20 min. Finally, the samples were transferred to absolute acetone for 20 min. The samples were placed in a 1:1 mixture of absolute acetone and the final Spurr resin mixture for 1 h at room temperature and then transferred to a 1:3 mixture of absolute acetone and the final resin mixture for 3 h and finally to a Spurr resin mixture overnight. After that, the samples were placed in an Eppendorf tube containing Spurr resin and heated at 70 °C for more than 9 h. The samples were then sectioned with a Leica EM UC7 ultratome, and the sections were stained with uranyl acetate and alkaline lead citrate for 10 min, respectively. A Hitachi Model H-7650 transmission electron microscope was used to visualize each sample. The numbers of IVs were measured from areas around rough canals (10 μm × 10 μm) in the micrographs of the *PTCDS*-knockdown and control wasps. Five different images were examined for each group. The numbers of MSEVs were measured from three randomly selected areas (2 μm × 2 μm) in the micrographs of the *PTCDS*-knockdown and control wasps. Three different images were examined for each group.

The immunoelectron microscopy (EM) experiments of the venom protein Lar were performed to confirm that IVs are responsible for the transportation of venom proteins in venom gland secretory cells. Approximately 30 venom glands from 3-day-old wild-type *L. heterotoma* female adults were treated with the primary Lar-specific antibody (1:20) and stained with goat anti-rabbit secondary antibodies (1:100) linked to 10 nm gold beads (Sigma). The negative control was treated with the non-immunogenicity serum instead of the primary Lar-specific antibody. A Hitachi Model H-7650 transmission electron microscope was used to visualize each sample.

## Data analysis and statistics

All statistical analyses were performed in GraphPad Prism version 8.0 (GraphPad Software) and SPSS 26.0 (IBM). The normal distribution of all the data was checked using the Shapiro–Wilk test, and the homogeneity of variance of all the data was checked via the Fligner–Killeen test. For comparisons between two groups, two-tailed Student's *t* test was used when parametric assumptions and homogeneity of variances were met, Welch's *t* test was used when parametric assumptions were met but heterogeneity of variances was observed, and the Mann–Whitney *U* test was used for nonparametric data. For multiple group comparisons, the Kruskal–Wallis test with Dunn's multiple comparisons test was used for nonparametric data. Details of the statistical analysis were provided in the figure legends. The data represent the mean ± standard error of the mean (SEM). Significance values were indicated as ns: not significant; $*P < 0.05$, $**P < 0.01$, and $***P < 0.001$.

## Data availability

Raw RNA transcriptome data were deposited in NCBI GenBank with Accession numbers: PRJNA1178739, PRJNA1178743, PRJNA1006516, PRJNA624738, PRJNA1379232. The MS proteome data were deposited to the ProteomeXchange Consortium via iProX partner repository with the dataset identifier PXD048783 (https://www.iprox.cn//page/project.html?id=IPX0008043000).

The source data of this paper are collected in the following database record: biostudies:S-SCDT-10_1038-S44318-026-00702-6.

## Peer review information

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

## Acknowledgements

We thank Chao Liu for helping make the proposed model figure. JH was supported by the National Natural Science Foundation of China (32325044, 32172467) and Zhejiang Provincial Natural Science Foundation of China (LZ23C140003). XXS was supported by the National Key R&D Program of China (2022YFD1401600), the National Science Foundation for Distinguished Young Scholars of Zhejiang Province (LR23C140001), and the National Natural Science Foundation of China (32071665). JC was supported by the National Natural Science Foundation of China (32202375). Research in Antonis Rokas's laboratory was supported by grants from the National Science Foundation (DEB-2110404), the National Institutes of Health/National Institute of Allergy and Infectious Diseases (R01 AI153356), and the Burroughs Wellcome Fund.

## Author contributions

**Zhiguo Liu**: Data curation; Validation; Investigation; Visualization; Methodology; Writing—original draft; Writing—review and editing. **Mei Tao**: Software; Investigation; Visualization; Methodology. **Zixuan Xu**: Resources; Investigation; Methodology. **Junwei Zhang**: Validation; Investigation; Visualization. **Yang Li**: Software; Validation; Methodology. **Zhi Dong**: Resources; Methodology. **Qichao Zhang**: Investigation. **Lan Pang**: Validation; Investigation. **Yifeng Sheng**: Validation; Investigation. **Yueqi Lu**: Validation; Investigation. **Ting Feng**: Validation; Investigation. **Wenqi Shi**: Validation; Investigation. **Longtao Yu**: Validation; Investigation. **Antonis Rokas**: Funding acquisition; Writing—original draft. **Jiani Chen**: Supervision; Funding acquisition; Investigation; Visualization; Writing—review and editing. **Xing-Xing Shen**: Supervision; Funding acquisition; Writing—original draft. **Jianhua Huang**: Conceptualization; Supervision; Funding acquisition; Writing—original draft; Project administration; Writing—review and editing.

Source data underlying figure panels in this paper may have individual authorship assigned. Where available, figure panel/source data authorship is listed in the following database record: biostudies:S-SCDT-10_1038-S44318-026-00702-6.

## Disclosure and competing interests statement

AR is a scientific consultant for LifeMine Therapeutics, Inc. The authors declare no competing interests.

# Expanded View Figures

**Figure EV1.  A phylogenetic tree of a gene family with putative HGT event3.**

The phylogeny was reconstructed (using a 90% identity-filtered sequence set) using IQ-TREE under the best-fit model selected by the -m MFP option. The resulting tree was midpoint-rooted, and branch support is shown as ultrafast bootstrap values (only values <95% are indicated near the internodes). Red branches indicate *L. heterotoma* and *L. syphax*, while cyan branches indicate Bacteria.

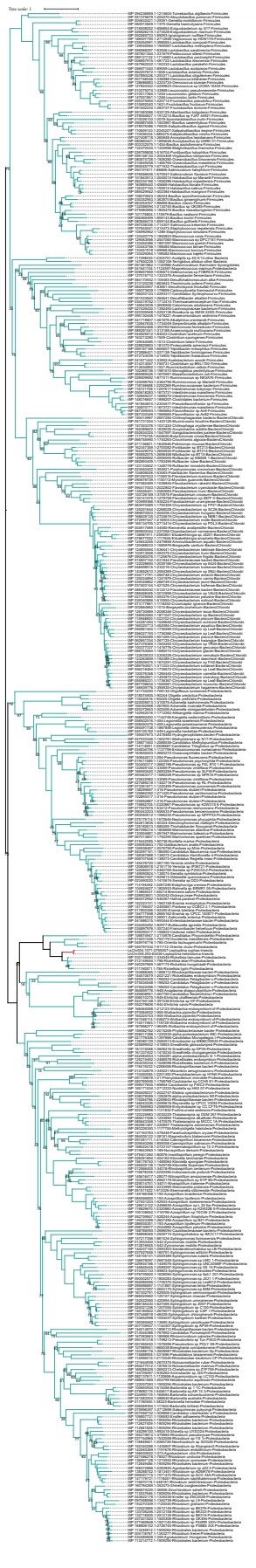

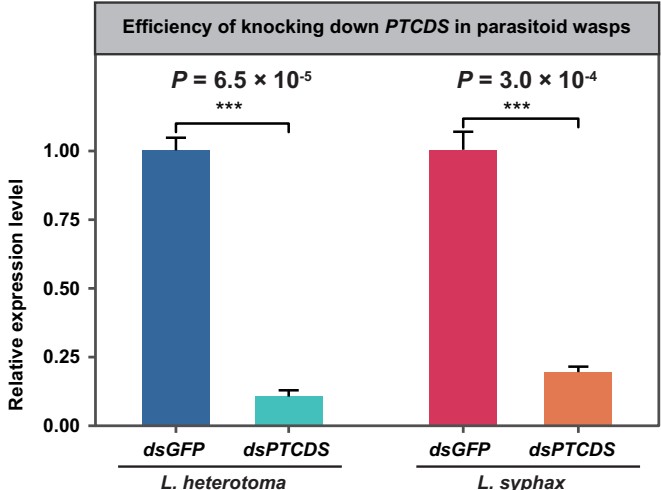

**Figure EV2. Efficiency of knocking down *PTCDS* in parasitoid wasps *L. heterotoma* and *L. syphax*.**

*dsGFP* was used as a control. Whole body of 3-day-old female adult *L. heterotoma* and 9-day-old female adult *L. syphax* was used to examine the gene repression. Three replicates were performed for each treatment. Data are presented as mean ± SEM. Significance was determined by two-tailed Student's *t* test (***$P < 0.001$). Source data are available online for this figure.

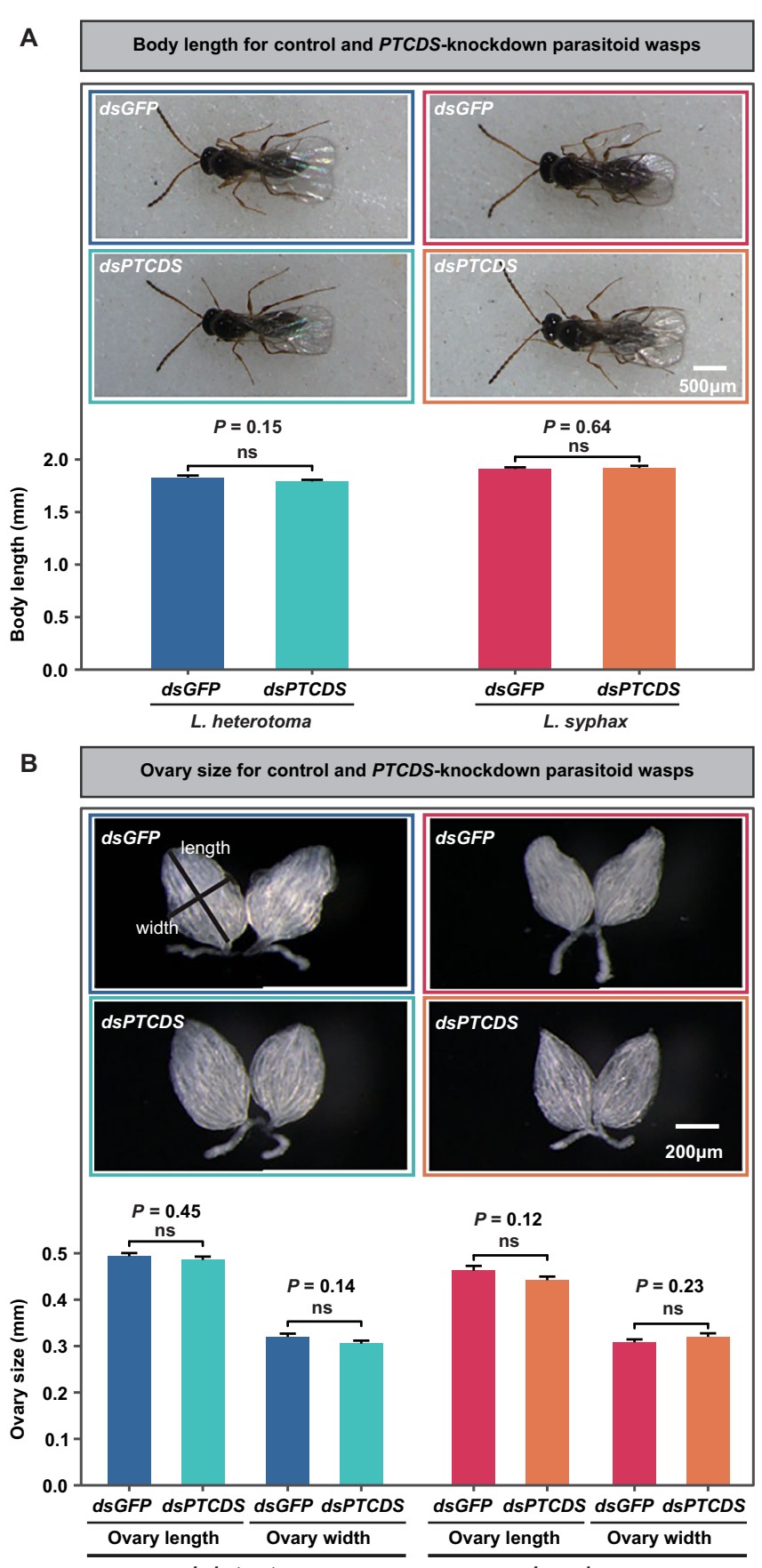

◄ **Figure EV3. Comparison of developmental phenotypes between control and *PTCDS*-knockdown parasitoid wasps.**

3-day-old female adult *L. heterotoma* ($n = 30$) and 9-day-old female adult *L. syphax* ($n = 25$) were used to examine developmental phenotypes, respectively. *dsGFP* was used as a control. (A) Effects of *PTCDS* on parasitoid wasp body length. (B) Effects of *PTCDS* on parasitoid wasp ovary size (length and width). Note that the length and width sizes are the average values of the pair of ovaries from each individual. Data are presented as mean ± SEM. Significance was determined by two-tailed Student's *t* test (ns, not significant). Source data are available online for this figure.

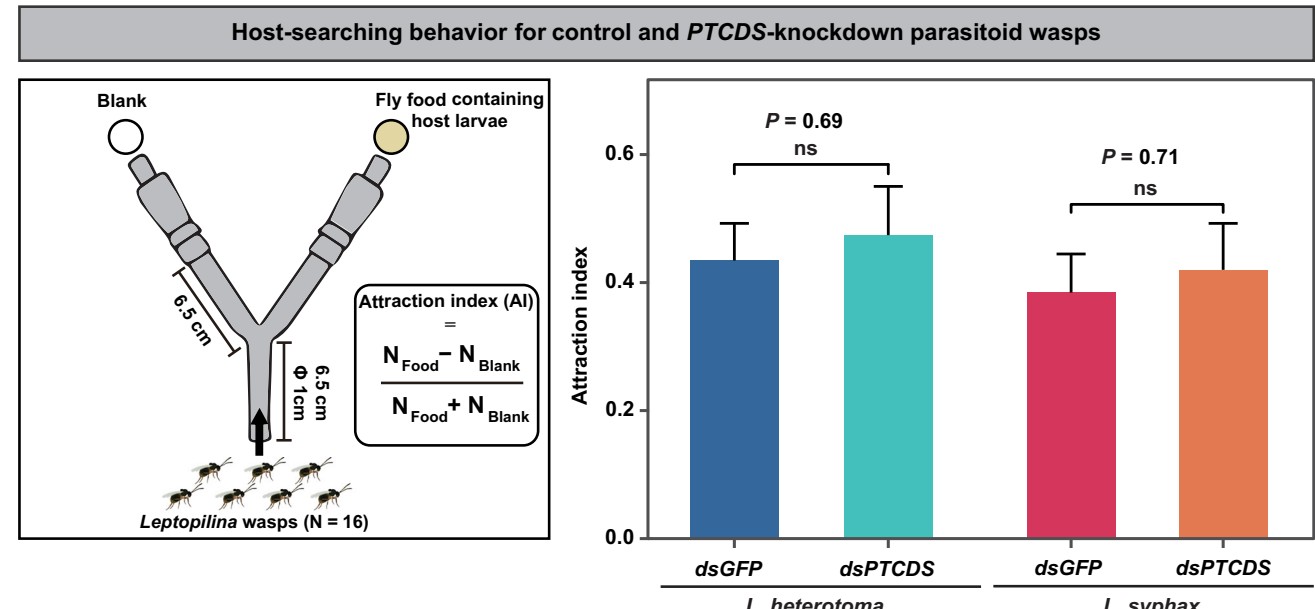

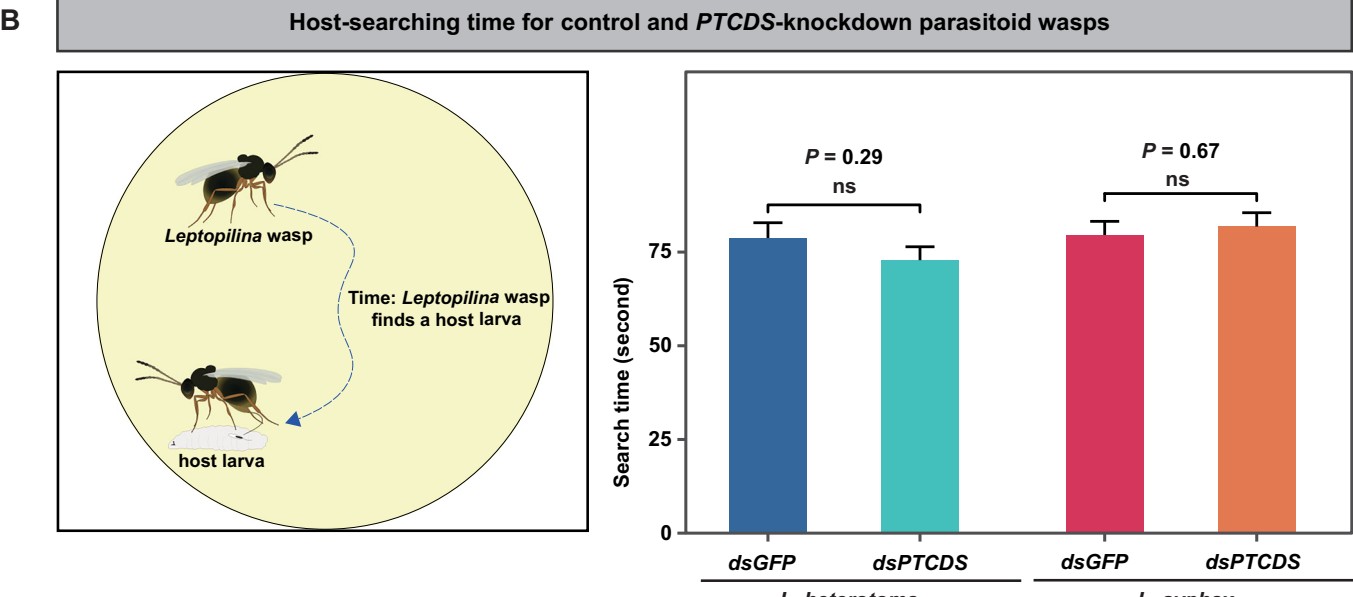

**Figure EV4.** Comparison of host-searching behaviors for control and *PTCDS*-knockdown parasitoid wasps.

(**A**) Effects of *PTCDS* on host searching ability for parasitoid wasps using Y-tube assay. The left panel is a schematic diagram of Y-tube behavioral assay for monitoring the host-searching behavior (Attraction Index, AI). The right panel is the summary of AI values for control and *PTCDS*-knockdown parasitoid wasps. *dsGFP* was used as a control. Six replicates were performed for each treatment. Data are presented as mean ± SEM. Significance was determined by two-tailed Student's *t* test (ns, not significant). (**B**) Effects of *PTCDS* on the time of searching host larvae for parasitoid wasps. The left panel is a schematic diagram of an assay for monitoring time that a wasp found a host larva. The right panel is the summary of the time of searching host. *dsGFP* was used as a control. Fifteen replicates were performed for each treatment. Data are presented as mean ± SEM. Significance was determined by two-tailed Student's *t* test (ns, not significant). Source data are available online for this figure.

## Oviposition rates for control and *PTCDS*-knockdown parasitoid wasps on larvae of six different *Drosophila* species

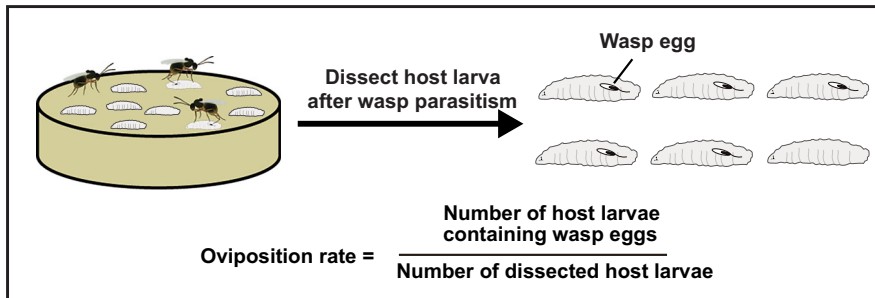

Oviposition rate = $\dfrac{\text{Number of host larvae containing wasp eggs}}{\text{Number of dissected host larvae}}$

**Figure EV5. Comparison of oviposition rate between control and *PTCDS*-knockdown parasitoid wasps.**

The upper panel is a schematic diagram for examining the oviposition rate of parasitoid wasps. The below panel is the summary of oviposition rates of parasitoid wasps that parasitized for each of six different *Drosophila* host larvae. *dsGFP* was used as a control. The ratio of female wasps to host larvae is 3:360 and 3:180 for 3-day-old *L. heterotoma* and 9-day-old *L. syphax*, respectively. Three replicates were performed for each treatment. Data are presented as mean ± SEM. Significance was determined by two-tailed Student's *t* test (ns, not significant). Source data are available online for this figure.

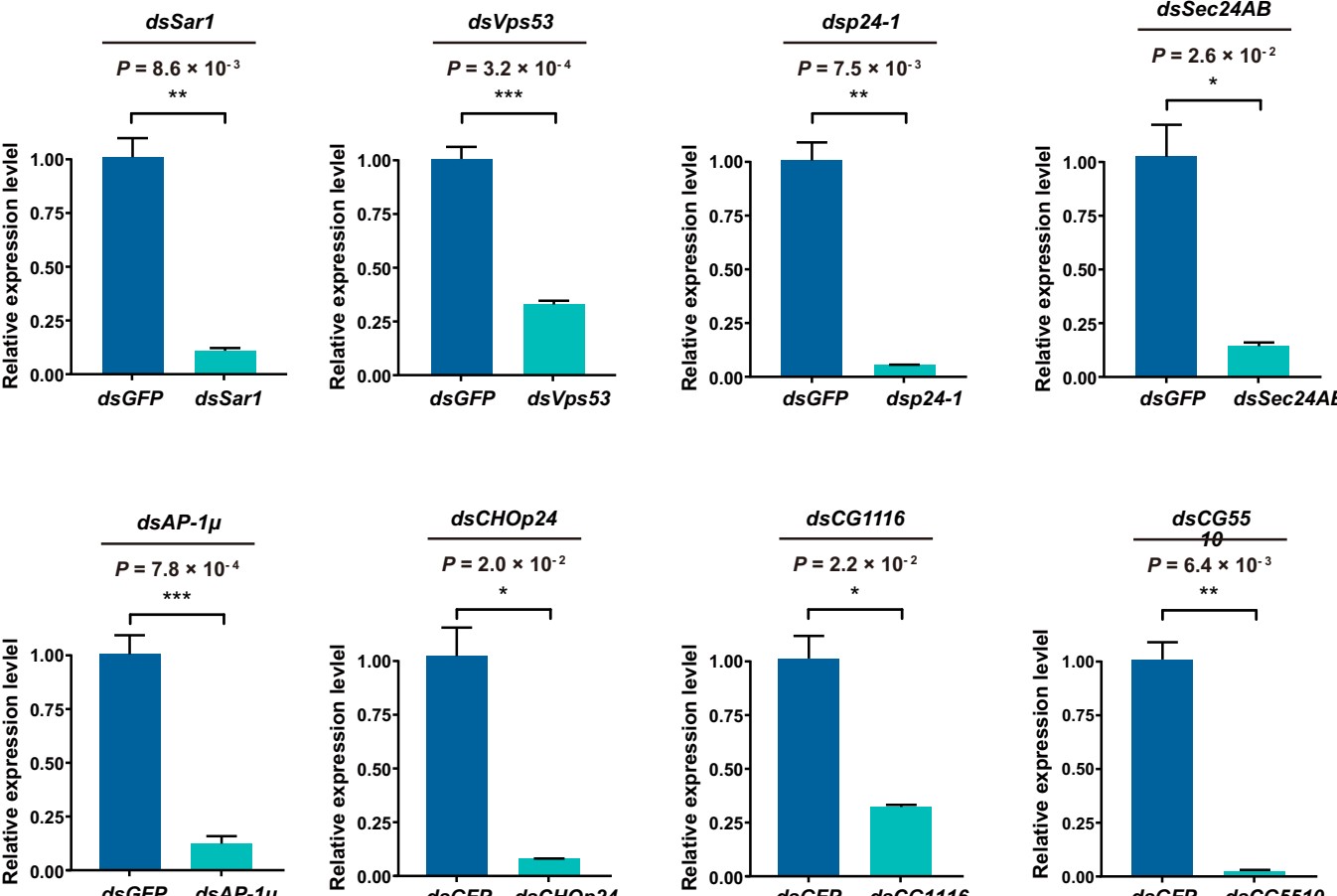

**Figure EV6.  Efficiency of knocking down eight protein transport-related genes in parasitoid wasp *L. heterotoma*.**

*dsGFP* was used as a control. Whole body of 3-day-old female adult *L. heterotoma* was used to examine the gene repression. Three replicates were performed for each treatment. Data are presented as mean ± SEM. Statistical analysis was performed using two-tailed Student's *t* test when parametric assumptions and homogeneity of variances were met, and Welch's *t* test was used to determine significance when parametric assumptions were met but heterogeneity of variances was observed ($*P < 0.05$; $**P < 0.01$; $***P < 0.001$). Source data are available online for this figure.

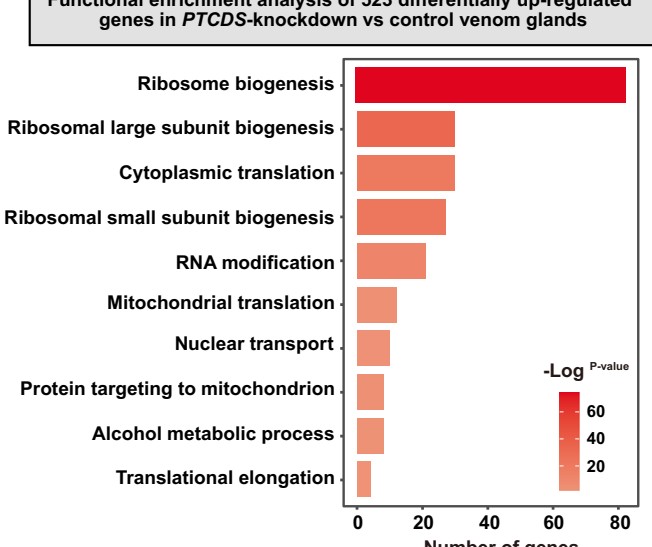

**Figure EV7. Functional enrichment analysis of 523 differentially upregulated genes.**

We conducted functional enrichment analysis of 523 differentially upregulated genes between *PTCDS*-knockdown and control venom glands of 3-day-old female adult *L. heterotoma*. Statistical significance was assessed using the hypergeometric test, *P* value < 0.01.

**Morphology of MSEVs in the venom reservoir of control and *PTCDS*-knockdown *L. heterotoma***

Venom gland

Venom reservoir

ds*GFP*

Venom gland

Venom reservoir

ds*PTCDS*

**Figure EV8.   *PTCDS* knockdown in *L. heterotoma* did not alter MSEV morphology.**

Comparison of transmission electron micrographs (TEMs) of MSEV morphology in the anterior and posterior regions of venom reservoirs from the control and *PTCDS*-knockdown 3-day-old female adult *L. heterotoma*.

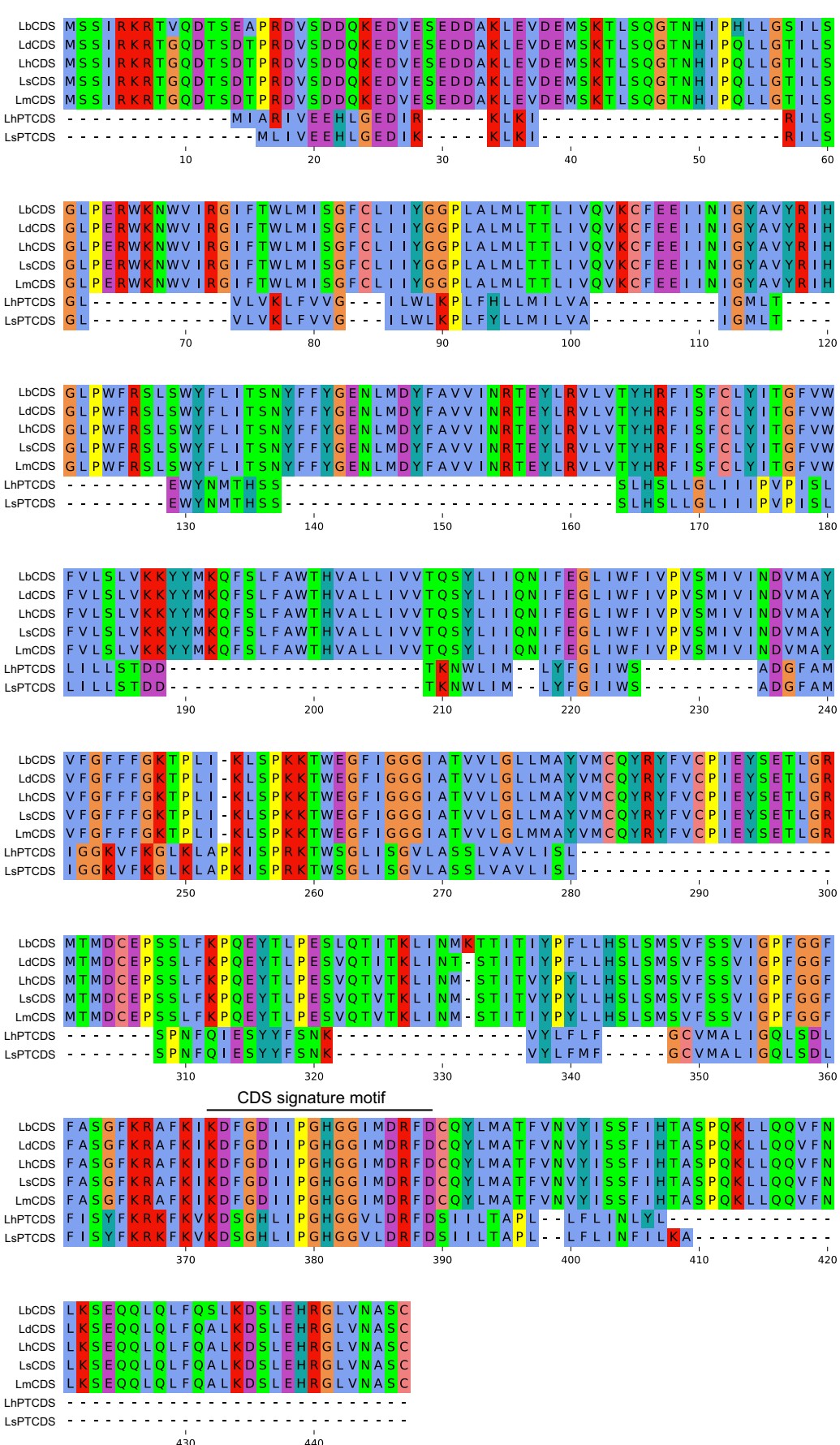

**Figure EV9. Comparison of eukaryotic-type and prokaryotic-type CDS protein sequences.**

The eukaryotic-type CDS proteins of *L. boulardi* (LbCDS), *L. drosophilae* (LdCDS), *L. heterotoma* (LhCDS), *L. syphax* (LsCDS) and *L. myrica* (LmCDS) exhibited high sequence conservation, sharing 97% amino acid identity. Although the eukaryotic-type CDS shares only ~27% amino acid identity with the PTCDS in *L. heterotoma* and *L. syphax*, all homologs contain a CDP-diacylglycerol synthetase domain, including a conserved CDS signature motif (marked by a black line).

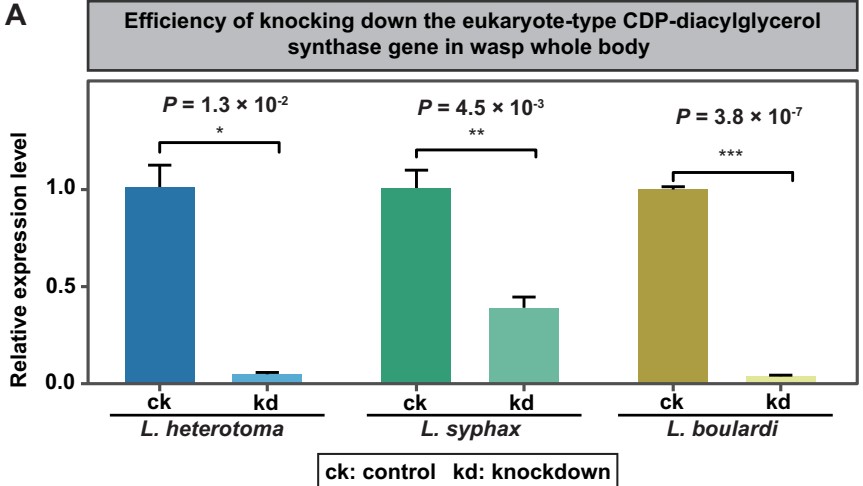

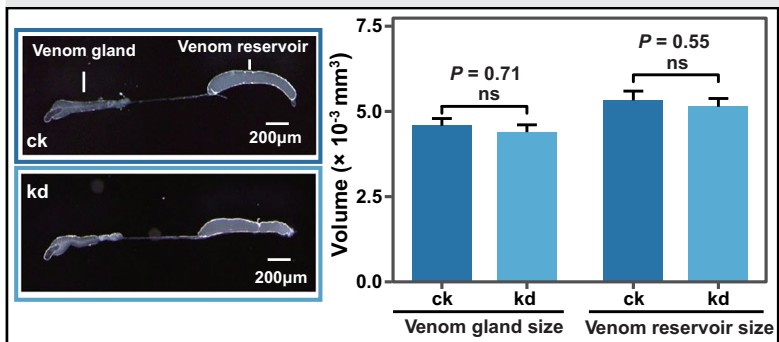

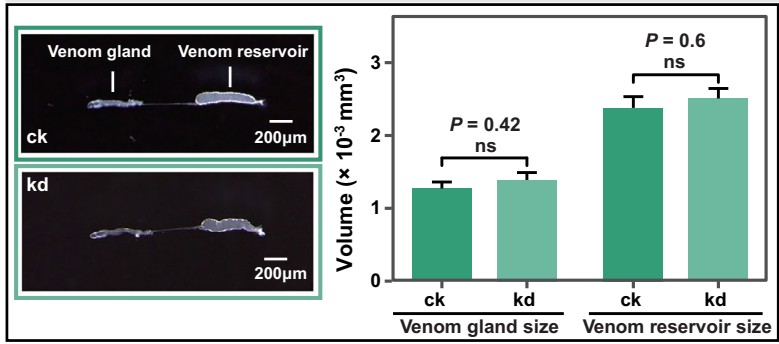

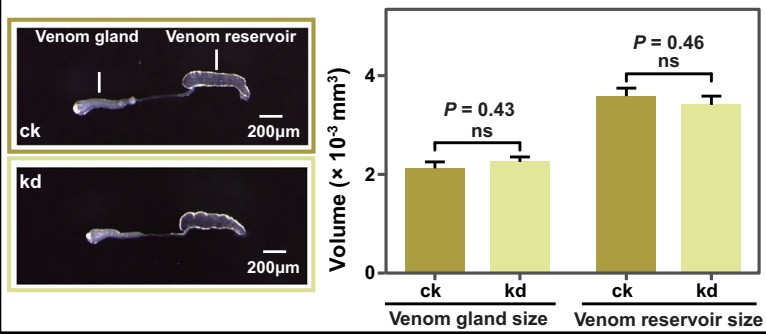

◀ **Figure EV10.  Effects of the eukaryote-type CDP-diacylglycerol synthase gene on venom gland size and venom reservoir size of parasitoid wasps.**

Although *L. heterotoma* and *L. syphax* horizontally acquired the prokaryote-type CDP-diacylglycerol synthase gene *(PTCDS)* from the bacterial family Rickettsiaceae, we found that all 5 parasitoid wasps (*L. heterotoma, L. syphax, L. drosophilae, L. boulardi*, and *L. myrica*) contain the eukaryote-type CDP-diacylglycerol synthase gene. The eukaryote-type CDP-diacylglycerol synthase gene shares ~20% amino acid identity with the prokaryote-type CDP-diacylglycerol synthase gene *(PTCDS)* in *L. heterotoma* and *L. syphax*. To examine the effects of eukaryote-type CDP-diacylglycerol synthase gene on venom system, we used RNAi to knock down the expression level of the eukaryote-type CDP-diacylglycerol synthase gene in *L. heterotoma, L. syphax*, and *L. boulardi*, respectively. (**A**) Efficiency of knocking down eukaryote-type CDP-diacylglycerol synthase gene in *L. heterotoma, L. syphax*, and *L. boulardi*. Three replicates were performed for each treatment. Data are presented as mean ± SEM. Statistical analysis was performed using two-tailed Student's *t* test when parametric assumptions and homogeneity of variances were met, and Welch's *t* test was used to determine significance when parametric assumptions were met but heterogeneity of variances was observed (*$P < 0.05$; **$P < 0.01$; ***$P < 0.001$). (**B**) Effects of knocking down the eukaryote-type CDP-diacylglycerol synthase gene on venom gland size and venom reservoir size in *L. heterotoma, L. syphax*, and *L. boulardi*. Female adult wasps *L. heterotoma* ($n = 30$), *L. syphax* ($n = 23$), and *L. boulardi* ($n = 30$) were used to examine their venom gland sizes and venom reservoir sizes, respectively. Data are presented as mean ± SEM. Statistical analysis was performed using two-tailed Student's *t* test when parametric assumptions and homogeneity of variances were met, Welch's *t* test was used to determine significance when parametric assumptions were met but heterogeneity of variances was observed, and the Mann–Whitney *U* test was used to determine significance when experiments required nonparametric statistical tests (ns, not significant). Source data are available online for this figure.

