## [Peer Review File · The EMBO Journal]

A bacterial gene acquired by parasitoid wasps contributes to venom secretion against host defence

Jianhua Huang, Zhiguo Liu, Mei Tao, Zixuan Xu, Junwei Zhang, Yang Li, Zhi Dong, Qichao Zhang, Lan Pang, Yifeng Sheng, Yueqi Lu, Ting Feng, Wenqi Shi, Longtao Yu, Antonis Rokas, Jiani Chen, and Xing-Xing Shen

Corresponding authors: Jianhua Huang (jhhuang@zju.edu.cn) , Xing-Xing Shen (xingxingshen@zju.edu.cn), Jiani Chen (jnchen@zju.edu.cn)

Review Timeline:

Submission Date:	6th Sep 25
Editorial Decision:	2nd Oct 25
Revision Received:	12th Nov 25
Editorial Decision:	9th Dec 25
Revision Received:	12th Dec 25
Accepted:	2nd Jan 26

Editor: Yehu Moran

Transaction Report:

Dear Prof. Huang,

Thank you for submitting your manuscript for consideration by the EMBO Journal. It has now been seen by three referees whose comments are shown below.

Given the referees' positive recommendations, I would like to invite you to submit a revised version of the manuscript, addressing the comments of all three reviewers. I should add that it is EMBO Journal policy to allow only a single round of revision, and acceptance of your manuscript will therefore depend on the completeness of your responses in this revised version.

Based on past experience, I strongly suggest that after consultation with your co-authors you will prepare a revision plan and share it with me via email in the next few weeks so I can comment on it. This can help in making the revision process much smoother and help in setting practical expectations from both the author's and editor's sides, increasing the chances of eventual acceptance of the article.

When preparing your point-by-point letter of response to the referees' comments, please bear in mind that this will form part of the Review Process File, and will therefore be available online to the community. For more details on our Transparent Editorial Process, please visit our website: <https://www.embopress.org/page/journal/14602075/authorguide#transparentprocess>

Thank you for the opportunity to consider your work for publication. I look forward to your revision.

Yours sincerely,

Yehu Moran
Academic Editor
The EMBO Journal

- a Reagents and Tools Table as part of the Methods section, which can be downloaded from our author guidelines

We realize that it is difficult to revise to a specific deadline. In the interest of protecting the conceptual advance provided by the work, we recommend a revision within 3 months (31st Dec 2025). Please discuss the revision progress ahead of this time with the editor if you require more time to complete the revisions.

Referee #1:

This is a fascinating study that provide insight into the role of HGT in the evolutionary arms race between parasitoids and their hosts. The authors identified 19 genes across five species of *Leptopilina* that originated from five HGT events. They then investigate the role of one of these genes using an impressive combination of experiments, including combination of gene knock-down and a wide range of behavioural and offspring survival assays. The results show that this HGT event has had a positive impact on fitness of offspring through increase survival through development in their host. As such the findings of the paper are highly interesting regardless of my concerns below regarding the findings on the mechanisms by which this increase in fitness is achieved.

PTCDS knockdown resulted in smaller venom reservoirs but the difference in venom reservoir was as far as I understand not accounted for when measuring venom concentration. That means that since the concentration of venom was determined by diluting the venom from the same number of venom reservoirs (i.e. different volumes of venom) in the same volume of water, the difference in sizes of the venom reservoirs between control and PTCDS knockdown is expected to result in differences in protein concentration estimates because the venom from the smaller venom reservoirs will be more diluted. Indeed, the differences between venom reservoir sizes and protein concentrations seem proportional. PTCDS does therefore not necessarily affect protein concentration of the venom.

Given that the venom reservoirs were not depleted prior to testing expression, the venom glands supplying toxins to the small and large venom reservoirs could also be in different states: actively replenishing venom with toxins in the wasps with larger venom reservoirs (and hence greater venom storage capacity that takes longer to replenish with toxins) versus and no longer actively replenishing venom with toxins in the wasps with smaller reservoirs (i.e. more quickly fully stocked with toxins). Since the concentration of toxins probably involves some kind of feedback mechanism to the venom gland to limit toxin production (e.g., in order to avoid i) excessive cost of venom production and/or ii) over-concentration of venom proteins leading to precipitation), these differences could very well explain the differential expression patterns and differences in intracellular vesicles between the control and knock-down groups.

On the other hand, I agree that the over-expression of PTCDS in the venom gland compared to the venom reservoir is indicative of an effect on toxin secretion into the venom reservoir. Depending on how the venom reservoir is filled, and whether this reflects not differences in capacity but differences in volume within similar limits of the venom reservoir, this role could be to either increase secretion rate of the venom gland or reduce the feedback signals from the venom reservoir (whatever they may be) that would in the absence of PTCDS result in reduced secretion rates as the venom reservoir is filled. I am not entirely sure how the authors could test this without having to repeat the differential expression experiment, but in my opinion it should at least be presented as one of the probable scenarios while toning down the current conclusions to reflect this possibility. The title and abstract should also be modified to reflect this uncertainty about the role of PTCDS.

For the identification of HGT genes, there are no data presented to support the conclusions either that they are of HGT origin or the number of HGT events that they represent, apart from a phylogeny of what seems like it may be a trimmed list of sequences. Since this is a central aspect of the paper I would like to see more evidence presented. Presenting data on identification of HGT genes would also provide readers with more detailed insight into these events, which would be very valuable to for example other ongoing and future work on HGT in this and other systems.

The methods and statistical outcomes lack some details that make it difficult to judge the results and conclusions.

- The RNAseq experiment used to investigate the expression of the HGT genes across tissues lacks details on the number of specimens used for each tissue, whether biological replicates were included, results of gene expression estimates, and variation across replicates.
- There are no details on how the LC-MS/MS data of trypsin digested venom was combined with the transcriptome or what settings and evidence and false positive thresholds were used.
- Statements of significance should also be accompanied by associated statistical details in the main text. Please also include replicates and variance estimates.
- The data from the immunoelectron microscopy (EM) experiments on the venom protein Lar that confirm that IVs are necessary to transport venom proteins in venom gland secretory cells (line 244) should be included.

Minor comments

- The names of the wasp genes in panel E of Fig. 1 are difficult to read. Seems like the coloured box is placed in from of the text?
- Line 94: "Involves two or more species" I think I understand what the authors mean in that these event occurred before the split of these species, but best modify this sentence to more clearly state this. Maybe using something like occurred in the common ancestors or occurred before the split.
- Line 110: delete "considerably" or replace with "highly"
- Line 135. Replace semicolon with a period And capitalize "the".
- Line 208 consider stating how many fold higher expression as opposed to "extreme"

Referee #2:

This is a very nice paper, identifying a gene, CDP-diacylglycerol synthase (PTCDS), which is uniquely present in two of the wasp species that parasitise drosophilid flies. The gene was recently (relatively speaking) acquired by the ancestor of these wasps, presumably by horizontal gene transfer from Rickettsia or a closely related bacterium. Knock-down of PTCDS reduces the parasitic success of the wasp larvae, which become more efficiently encapsulated by the host. This is correlated to a reduction of venom production in the PTCDS-deficient female wasp.

The evolutionary importance of horizontal gene transfer has only recently become appreciated, at least for eukaryotes, and little is known about the mechanisms involved. The story presented in this manuscript is particularly interesting, as it documents a function and a likely selective advantage for a horizontally transferred gene. The results are of interest for the understanding of evolution in general, and specifically for the co-evolution of parasites and hosts. The conclusions are well supported by experimental data.

Comments:

1. The paper is well written and well referenced, but I miss an introductory background about the physiological role of CDP-diacylglycerol synthases.
2. One obvious question comes up when you read this paper: If bacteria-derived PTCDS is required for efficient secretion of venom in these two wasp species, how do all other wasps manage? I don't expect a conclusive answer to this complex question, but the authors have made a first attempt to address it by testing the role of the endogenous, eukaryotic CDP-diacylglycerol synthase, finding no role of that gene as it seems. That result is briefly described in the discussion only. While negative, I suggest that this result could be better highlighted and transferred from the supplement to the results section.
3. Is PTCDS present among the proteins found in the venom (Table S9) or is it located in the venom gland only?

Minor point:

lines 275-276: "found that found that"

Referee #3:

It was a pleasure to read this well-designed and elegant study. It describes the function of a horizontally transferred bacterial enzyme in a parasitoid wasp, in which it regulates the secretion of venom. The authors show by knockdown experiments that this enzyme plays a central role in venom secretion from the venom gland into the storage reservoir. Knocking down the enzyme decreases the amount of venom in the venom reservoir, which inhibits the wasp's ability to combat the host's immune defences. This shines a unique light on the adaptive value of horizontally transferred genes for parasitoid success. The work is beautifully done, well documented, nicely illustrated, and clearly presented. I congratulate the authors on this fine piece of work.

I have no major concerns. Indeed, the only minor suggestion I have is to replace the word "confer" in line 36 with "use".

Referee #1:

This is a fascinating study that provide insight into the role of HGT in the evolutionary arms race between parasitoids and their hosts. The authors identified 19 genes across five species of *Leptopilina* that originated from five HGT events. They then investigate the role of one of these genes using an impressive combination of experiments, including combination of gene knock-down and a wide range of behavioural and offspring survival assays. The results show that this HGT event has had a positive impact on fitness of offspring through increase survival through development in their host. As such the findings of the paper are highly interesting regardless of my concerns below regarding the findings on the mechanisms by which this increase in fitness is achieved.

AUTHORS' RESPONSE: We sincerely thank the Reviewer for the positive assessment of our work as "a fascinating study" and "highly interesting," and for the thoughtful comments that help us improve the manuscript.

PTCDS knockdown resulted in smaller venom reservoirs but the difference in venom reservoir was as far as I understand not accounted for when measuring venom concentration. That means that since the concentration of venom was determined by diluting the venom from the same number of venom reservoirs (i.e. different volumes of venom) in the same volume of water, the difference in sizes of the venom reservoirs between control and PTCDS knockdown is expected to result in differences in protein concentration estimates because the venom from the smaller venom reservoirs will be more diluted. Indeed, the differences between venom reservoir sizes and protein concentrations seem proportional. PTCDS does therefore not necessarily affect protein concentration of the venom.

AUTHORS' RESPONSE: We thank the reviewer for this critical point, which allows us to clarify our methodology and results. We agree with the reviewer that PTCDS knockdown does not necessarily affect the concentration of proteins within the venom fluid itself. However, our central claim is that it reduces the total amount of venom proteins per venom reservoir, not the concentration. For the SDS-PAGE analysis, total protein was extracted from a pool of 20 venom reservoirs per biological replicate and dissolved in a fixed volume of buffer to visualize the protein profile. For the quantitative BCA assay, total protein was extracted from one individual venom reservoir per biological replicate and dissolved in a fixed volume of buffer. Therefore, the reported values (Fig. 3B) represent the absolute total yield from the organ, independent of its concentration. To prevent any misunderstanding, we have revised the text to consistently use "total venom protein amount" (lines 214, 217, and 234) and have updated the Y-axis title in Fig. 3C from "Relative concentration of venom protein Lar" to "Relative level of venom protein Lar" and have also revised the relevant section in the Methods accordingly (lines 631-632).

Given that the venom reservoirs were not depleted prior to testing expression, the venom glands supplying toxins to the small and large venom reservoirs could also be in different states: actively replenishing venom with toxins in the wasps with larger venom reservoirs (and hence greater venom storage capacity that takes longer to replenish with toxins) versus and no longer actively replenishing venom with toxins in the wasps with smaller reservoirs (i.e. more quickly fully stocked with toxins). Since the concentration of toxins probably involves some kind of feedback mechanism to the venom gland to limit toxin production (e.g., in order to avoid i) excessive cost of venom

production and/or ii) over-concentration of venom proteins leading to precipitation), these differences could very well explain the differential expression patterns and differences in intracellular vesicles between the control and knock-down groups.

On the other hand, I agree that the over-expression of *PTCDS* in the venom gland compared to the venom reservoir is indicative of an effect on toxin secretion into the venom reservoir. Depending on how the venom reservoir is filled, and whether this reflects not differences in capacity but differences in volume within similar limits of the venom reservoir, this role could be to either increase secretion rate of the venom gland or reduce the feedback signals from the venom reservoir (whatever they may be) that would in the absence of *PTCDS* result in reduced secretion rates as the venom reservoir is filled. I am not entirely sure how the authors could test this without having to repeat the differential expression experiment, but in my opinion it should at least be presented as one of the probable scenarios while toning down the current conclusions to reflect this possibility. The title and abstract should also be modified to reflect this uncertainty about the role of *PTCDS*.

AUTHORS' RESPONSE: We thank the reviewer for proposing this interesting alternative scenario. We agree that feedback from the venom reservoir is a plausible biological mechanism in general. However, the expression profiles strongly support a primary role for *PTCDS* acting within the venom gland to influence secretion. Our RNA-seq (Fig. 1D) and qRT-PCR data (Fig. 4A) unambiguously show that *PTCDS* is expressed at an extremely high level in the venom gland compared to all other tissues, including the venom reservoir. This specific expression pattern is most consistent with a function intrinsic to the secretory process within the gland cells. While we cannot fully rule out the existence of a secondary feedback loop from the reservoir, we have revised the manuscript to acknowledge the reviewer's point. Specifically, we have:

- 1) Added a paragraph in the Discussion: "While our data strongly support a model where *PTCDS* functions within the venom gland cells to regulate vesicle-mediated secretion, we cannot rule out the possibility that the reduced venom reservoir volume in knockdown wasps might also influence the venom gland's activity through yet-unknown feedback mechanisms" (lines 390-394).
- 2) Modified the title to "A bacterial gene acquired by parasitoid wasps contributes to venom secretion against host defence".
- 3) Toned down corresponding conclusions in the Abstract to reflect this possibility (lines 29, 30, and 36).

For the identification of HGT genes, there are no data presented to support the conclusions either that they are of HGT origin or the number of HGT events that they represent, apart from a phylogeny of what seems like it may be a trimmed list of sequences. Since this is a central aspect of the paper I would like to see more evidence presented. Presenting data on identification of HGT genes would also provide readers with more detailed insight into these events, which would be very valuable to for example other ongoing and future work on HGT in this and other systems.

AUTHORS' RESPONSE: We thank the reviewer for this critical comment regarding the evidence for HGT. In response, we have taken the following steps to provide full transparency and support for our conclusions:

- 1) Given its central role in our functional study, the detailed, untrimmed maximum likelihood tree for the two *PTCDS* genes (Event 3) is now provided as a new Figure EV1, as noted in the updated legend for Figure 1E (lines 968-969) and in the Materials and Methods section (lines 434-438).
- 2) The complete set of maximum likelihood phylogenetic trees for all 19 HGT candidates has been

deposited in the Figshare repository (<https://figshare.com/s/9e1c41f3c16b61db09a5>) and is publicly available (lines 434-438).

We believe these full phylogenetic datasets provide the necessary evidence for the HGT origins and the number of independent events, and we hope they will serve as a valuable resource for future studies on HGT in parasitoid wasps and other systems.

The methods and statistical outcomes lack some details that make it difficult to judge the results and conclusions.

- The RNAseq experiment used to investigate the expression of the HGT genes across tissues lacks details on the number of specimens used for each tissue, whether biological replicates were included, results of gene expression estimates, and variation across replicates.

AUTHORS' RESPONSE: We have provided the details of RNAseq experiment, including the number of specimens used for each tissue (lines 440-444) and results of gene expression estimates (Table EV2) and a detailed methods used for analysis (lines 454-459). While biological replicates were not sequenced for this specific tissue-expression survey, the consistent and high-level expression of *PTCDS* specifically in the venom gland was robustly validated by independent qRT-PCR experiments with biological replicates (n=3).

- There are no details on how the LC-MS/MS data of trypsin digested venom was combined with the transcriptome or what settings and evidence and false positive thresholds were used.

AUTHORS' RESPONSE: We have provided a detailed description in the Materials and Methods section "Identification of venom proteins via LC-MS/MS" regarding the database search settings, false discovery rate (FDR) threshold, and how the proteomic data was matched to the transcriptome-predicted venom proteins (lines 612-624).

- Statements of significance should also be accompanied by associated statistical details in the main text. Please also include replicates and variance estimates.

AUTHORS' RESPONSE: We thank the reviewer for this important comment. We have carefully re-examined our statistical analyses and revised the manuscript accordingly. While these revisions do not alter our original conclusions, they enhance the rigor and accuracy of our statistical reporting. We have now stated the statistical test, p-value, sample size (n), and variance estimates (mean \pm SEM) for all significant statements both within the main text and the corresponding figure legends (lines 129-141, 149-167, 173-176, 183-188, 199-204, 206-211, 222-223, 270-273, 276-280, 301, 305-306, 340-346, 986-987, 990-991, 996-998, 1005-1009, 1026-1028, 1041-1044, and 1068-1070). The statistical methods have also been provided in the Materials and Methods section (lines 701-711).

- The data from the immunoelectron microscopy (EM) experiments on the venom protein Lar that confirm that IVs are necessary to transport venom proteins in venom gland secretory cells (line 244) should be included.

AUTHORS' RESPONSE: It has been fixed (line 297).

Minor comments

- The names of the wasp genes in panel E of Fig. 1 are difficult to read. Seems like the coloured box

is placed in from of the text?

AUTHORS' RESPONSE: We have adjusted the figure to ensure all gene names are clearly readable (new Fig. 1E).

- Line 94: "Involves two or more species" I think I understand what the authors mean in that these event occurred before the split of these species, but best modify this sentence to more clearly state this. Maybe using something like occurred in the common ancestors or occurred before the split.

AUTHORS' RESPONSE: We have rephrased to: "three HGT events occurred in the common ancestor" (line 104).

-Line 110: delete "considerably" or replace with "highly"

AUTHORS' RESPONSE: We have replaced "considerably" with "highly" (line 120).

- Line 135. Replace semicolon with a period And capitalize "the".

AUTHORS' RESPONSE: We have replaced the semicolon with a period and capitalize "The" (line 164).

-Line 208 consider stating how many fold higher expression as opposed to "extreme"

AUTHORS' RESPONSE: We thank the reviewer for this suggestion. As recommended, we have replaced the subjective term "extreme" with the precise fold-change values in the text (lines 247-250).

Referee #2:

This is a very nice paper, identifying a gene, CDP-diacylglycerol synthase (PTCDS), which is uniquely present in two of the wasp species that parasitise drosophilid flies. The gene was recently (relatively speaking) acquired by the ancestor of these wasps, presumably by horizontal gene transfer from Rickettsia or a closely related bacterium. Knock-down of PTCDS reduces the parasitic success of the wasp larvae, which become more efficiently encapsulated by the host. This is correlated to a reduction of venom production in the PTCDS-deficient female wasp.

The evolutionary importance of horizontal gene transfer has only recently become appreciated, at least for eukaryotes, and little is known about the mechanisms involved. The story presented in this manuscript is particularly interesting, as it documents a function and a likely selective advantage for a horizontally transferred gene. The results are of interest for the understanding of evolution in general, and specifically for the co-evolution of parasites and hosts. The conclusions are well supported by experimental data.

AUTHORS' RESPONSE: We sincerely thank the Reviewer for the very positive and encouraging assessment of our manuscript as "a very nice paper" and for recognizing that "the results are of interest for the understanding of evolution in general."

Comments:

1. The paper is well written and well referenced, but I miss an introductory background about the physiological role of CDP-diacylglycerol synthases.

AUTHORS' RESPONSE: We agree that providing this context will be very helpful for a broad readership. We have added an introductory background about the fundamental role of CDP-diacylglycerol synthases in the revised manuscript (lines 319-329).

2. One obvious question comes up when you read this paper: If bacteria-derived PTCDS is required for efficient secretion of venom in these two wasp species, how do all other wasps manage? I don't expect a conclusive answer to this complex question, but the authors have made a first attempt to address it by testing the role of the endogenous, eukaryotic CDP-diacylglycerol synthase, finding no role of that gene as it seems. That result is briefly described in the discussion only. While negative, I suggest that this result could be better highlighted and transferred from the supplement to the results section.

AUTHORS' RESPONSE: We thank the reviewer for this excellent suggestion. As recommended, we have moved the description of the eukaryotic CDS knockdown experiment from the Discussion to a new, paragraph in the Results section, titled "The eukaryotic-type CDP-diacylglycerol synthase gene is not required for venom protein secretion in parasitoid wasps" (lines 317-352). This new section reports the negative results of its knockdown, which showed no effect on venom reservoir size and wasp offspring emergence rate (previously in Fig. S8 and Table S16). For clarity within the main narrative, the corresponding figure remains as a supplementary item (now referenced as Fig. EV10) but is directly cited in this new paragraph. This reorganization effectively highlights the functional specificity of the horizontally acquired *PTCDS*.

3. Is PTCDS present among the proteins found in the venom (Table S9) or is it located in the venom gland only?

AUTHORS' RESPONSE: This is a very insightful question. Our LC-MS/MS analysis of the venom reservoir proteome show that PTCDS is not present among the secreted venom proteins. We have now stated this in the manuscript: "Notably, PTCDS itself was not detected among the identified venom proteins" (line 230).

Minor point:

lines 275-276: "found that found that"

AUTHORS' RESPONSE: It has been corrected.

Referee #3:

It was a pleasure to read this well-designed and elegant study. It describes the function of a horizontally transferred bacterial enzyme in a parasitoid wasp, in which it regulates the secretion of venom. The authors show by knockdown experiments that this enzyme plays a central role in venom secretion from the venom gland into the storage reservoir. Knocking down the enzyme decreases the amount of venom in the venom reservoir, which inhibits the wasp's ability to combat the host's immune defences. This shines a unique light on the adaptive value of horizontally transferred genes for parasitoid success. The work is beautifully done, well documented, nicely illustrated, and clearly presented. I congratulate the authors on this fine piece of work.

AUTHORS' RESPONSE: We are sincerely thankful to the Reviewer for their exceptionally positive and generous assessment of our work.

I have no major concerns. Indeed, the only minor suggestion I have is to replace the word "confer" in line 36 with "use".

AUTHORS' RESPONSE: We thank the reviewer for this helpful suggestion. We have replaced "confer" with "use" (line 36).

Dear Prof. Huang,

Thank you for submitting your manuscript for consideration by the EMBO Journal. It has now been seen by three referees whose comments are enclosed. As you will see, all three referees express interest in your manuscript and are broadly in favour of publication, pending satisfactory minor revision. Please pay special attention to the comments of Referee #1 and amend the text according to their comments to address uncertainty. It is okay to state that some questions remain partially open and will require additional research in future work.

I would like to invite you to submit a revised version of the manuscript, addressing these comments.

In addition to the referee's comments, our editorial assistance team raised several technical issues that must be addressed and corrected prior to official acceptance.

When preparing your letter of response to the referees' comments, please bear in mind that this will form part of the Review Process File, and will therefore be available online to the community. There is no need to refer to the editorial assistance comments in the response letter. For more details on our Transparent Editorial Process, please review our Editorial Policies page: <https://link.springer.com/partners/embo-press/editorial-policies>

We generally allow three months as standard revision time. As a matter of policy, competing manuscripts published during this period will not negatively impact on our assessment of the conceptual advance presented by your study. However, in light of the minor nature of the remaining comments and the fact the requested revision is only textual, I would appreciate it if you can submit your revision by the end of the year.

Thank you for the opportunity to consider your work for publication. I look forward to your revision.

Yours sincerely,

Yehu Moran
Academic Editor
The EMBO Journal

Read our guidance for manuscript revisions and related editorial policies: <https://link.springer.com/journal/44318/submission-guidelines#cms-Revised-submissions>

<https://media.springernature.com/original/springer-cms/rest/v1/content/27825798/data/v1>

- a point-by-point response to the referees' comments, with a detailed description of the changes made (as a word file).
- a word file of the manuscript text.
- individual production quality figure files (one file per figure)
- a complete author checklist
- Expanded View files (replacing Supplementary Information)
- a Reagents and Tools Table as part of the Methods section

Please remember: Digital image enhancement is acceptable practice, as long as it accurately represents the original data and conforms to community standards. If a figure has been subjected to significant electronic manipulation, this must be noted in the figure legend or in the 'Methods' section. The editors reserve the right to request original versions of figures and the original images that were used to assemble the figure.

We realize that it is difficult to revise to a specific deadline. In the interest of protecting the conceptual advance provided by the work, we recommend a revision within 3 months (9th Mar 2026). Please discuss the revision progress ahead of this time with

the editor if you require more time to complete the revisions.

specific comments by the editorial assistance team

*AUTHOR CHECKLIST: Missing. Please provide the completed Author Checklist, which you can download from the top of the Author Guidelines.

*FIGURES IN SEPARATE FILES: Please upload the EV figures as separate figure files and place their legends in the manuscript text, after the main figure legends and under the heading "Expanded View Figure Legends".

*Keywords: please provide.

*FUNDING: Please merge with Acknowledgments, please enter the complete list of funders in our system in addition to the manuscript text. The list in our system will be directly linked to the funder's database for articles when they are published in PubMed; it is therefore important for this list in our system to be complete and accurate.

*Author Contributions: remove from manuscript text and ensure that the author contributions are correctly listed only in our system.

*DATASET EV LEGENDS: Please upload Tables EV1 - 17 as individual files, one per table, and keep the corresponding legends at the top of each page.

*SYNOPSIS IMAGE: not provided. Please provide according to our instructions.

*SYNOPSIS TEXT: not provided. Please provide according to our instructions.

Additional Notes:

- Remove the figures from the manuscript text file.

- Please update the order and the headings of the sections in the manuscript text to the following: Abstract / Keywords / Introduction / Results / Discussion / Methods / Data Availability / Acknowledgements / Disclosure and Competing Interests Statement / References / Figure Legends / Expanded View Figure Legends

- Please remove the list of suppl. materials from the manuscript text

- Data Availability Section:

Please note that the specific URLs for datasets (NCBI GenBank with Accession numbers: 926 SRR31143667, SRR31143668, SRR31143669, SRR31143670, SRR31143671, 927 SRR31143672, SRR30310313, SRR31143679, SRR31143680, SRR31143681, 928 SRR31143682, SRR31143683, SRR31143684, SRR13003863, SRR31156171, and 929 SRR31156172. The ProteomeXchange Consortium via iProX partner repository with the dataset identifier PXD048783.) are not provided in the data availability statement. Please correct.

- Figure legends:

1. Please note that the exact p values are not provided in the legends of figure 4d. Please provide.

2. Please indicate the statistical test used for data analysis in the legends of figures 4b; EV-7.

3. Please note that the scale bar needs to be defined for figure 4d.

Referee comments

Referee #1:

Thanks for addressing most of my comments. Please see below for some relatively minor comments in return.

Major comment 1: Protein concentration

Thanks for clarifying the point about concentration. However, that was exactly my point: these results do not show that "PTCDS is more likely to affect venom protein secretion than venom protein synthesis in venom gland cells," because the concentration seems to be the same in both sized reservoir, resulting in different total amounts between differently sized reservoirs.

I am not arguing that PTCDS has nothing to do with protein secretion in the venom glands, but these data do not necessarily support that conclusion. What the data show is that knocking out PTCDS seems to have an effect on the venom reservoir size and not on the properties of the venom stored there-in (including total amount per volume).

Major comment 2: PTCDS expression

I agree that the comparative expression data shows it is much more highly expressed in venom glands, but keep in mind these are probably resting state so are not necessarily actively secreting venom.

Also, secretion state is well known to change in venom glands as spent venom is repleted (i.e. venom storage is refilled), to the point that expression profiles completely change between actively regenerating and repleted venom glands. So different secretory states is to be expected when the total amount of venom to be replenished is different. The difference in the number of

IVs can also be attributed to this potential difference in secretion state.

I also agree that it is a bit of a puzzling finding, and to me the role of PTCDS still remains a bit of a mystery. Perhaps it actually affects venom reservoir size for example during development, I don't know, but rather than torpedoing the manuscript I suggest the authors should take these points into account. The work is still interesting even if the results are relatively inconclusive regarding the specific role of this gene.

Major comment 3: Accessibility of HGT data

Thank you for making the fasta files and trees available. It would also be very useful to include the sequence alignments used as this both provides valuable insights into protein evolution and influences tree topology.

Other comments: The RNAseq experiment

I agree that the additional use of qPCR, including triplicates, is a good validation of the PTCDS expression. The number of replicates (and lack thereof) is still important to mention in the manuscript. If anything to point out that despite the RNAseq replicates there were additional measures taken to confirm these results.

Referee #2:

The revised version of this manuscript deals with the questions raised by me and, I believe, the other reviewer in a fully satisfactory way.

Referee #3:

The authors have addressed my minor comments satisfactorily. Note that I have not assessed whether the authors have adequately addressed the comments of the other two reviewers.

Referee #1:

Thanks for addressing most of my comments. Please see below for some relatively minor comments in return.

AUTHORS' RESPONSE: We sincerely thank the reviewer for the positive feedback on our work. We also greatly appreciate the valuable comments, which have helped us improve the manuscript and make the descriptions of our findings more precise. We have revised the manuscript accordingly and provided a point-by-point response below.

Major comment 1: Protein concentration

Thanks for clarifying the point about concentration. However, that was exactly my point: these results do not show that "PTCDS is more likely to affect venom protein secretion than venom protein synthesis in venom gland cells," because the concentration seems to be the same in both sized reservoir, resulting in different total amounts between differently sized reservoirs.

I am not arguing that PTCDS has nothing to do with protein secretion in the venom glands, but these data do not necessarily support that conclusion. What the data show is that knocking out PTCDS seems to have an effect on the venom reservoir size and not on the properties of the venom stored there-in (including total amount per volume).

AUTHORS' RESPONSE: Thank you for this comment. We have revised the conclusion of Fig. 3 in the manuscript accordingly.

Original: *“Overall, these results show that knockdown of the HGT-acquired gene PTCDS reduced the total amount but did not change the composition of venom proteins in the venom reservoir, implying that PTCDS is more likely to affect venom protein secretion than venom protein synthesis in venom gland cells.”*

Revised: *“Overall, these results show that knockdown of the HGT-acquired gene PTCDS reduced the total amount but did not change the composition of venom proteins in the venom reservoir. This suggests a link between PTCDS function and the regulation of stored venom protein amount, which could be mediated by affecting protein secretion in the venom gland cells or by influencing venom reservoir size.”* (lines 238-242)

Major comment 2: PTCDS expression

I agree that the comparative expression data shows it is much more highly expressed in venom glands, but keep in mind these are probably resting state so are not necessarily actively secreting venom.

Also, secretion state is well known to change in venom glands as spent venom is repleted (i.e. venom storage is refilled), to the point that expression profiles completely change between actively regenerating and repleted venom glands. So different secretory states is to be expected when the total amount of venom to be replenished is different. The difference in the number of IVs can also be attributed to this potential difference in secretion state.

I also agree that it is a bit of a puzzling finding, and to me the role of PTCDS still remains a bit of

a mystery. Perhaps it actually affects venom reservoir size for example during development, I don't know, but rather than torpedoing the manuscript I suggest the authors should take these points into account. The work is still interesting even if the results are relatively inconclusive regarding the specific role of this gene.

AUTHORS' RESPONSE: Thank you for this comment. We agree that some questions remain partially open and will require additional research in future work. In the revised manuscript, we have revised the Discussion section to incorporate your insightful points.

Original: *“While our data strongly support a model where PTCDS functions within the venom gland cells to regulate vesicle-mediated secretion, we cannot rule out the possibility that the reduced venom reservoir size in knockdown parasitoid wasps might also influence the venom gland's activity through yet-unknown feedback mechanisms.”*

Revised: *“While our data suggest a model in which PTCDS functions within the venom gland cells to regulate vesicle-mediated secretion, we cannot rule out the alternative possibility that PTCDS regulates venom reservoir size, which in turn could affect the amount of stored venom protein, possibly through yet-unknown mechanisms that influence the venom gland's activity. Furthermore, as the stored venom protein in venom reservoir is dynamically replenished after use, the secretion state of the gland is not static. Therefore, future work is needed to elucidate the precise role of PTCDS in these dynamic processes, specifically whether it contributes to the secretory machinery, modulates venom reservoir development, or both.”* (lines 393-401)

Major comment 3: Accessibility of HGT data

Thank you for making the fasta files and trees available. It would also be very useful to include the sequence alignments used as this both provides valuable insights into protein evolution and influences tree topology.

AUTHORS' RESPONSE: Thank you. The sequence alignments have now been uploaded and are publicly available via the Figshare repository (<https://figshare.com/s/801551d66a83eaaafde9>). (line 444)

Other comments: The RNAseq experiment

I agree that the additional use of qPCR, including triplicates, is a good validation of the PTCDS expression. The number of replicates (and lack thereof) is still important to mention in the manuscript. If anything to point out that despite the RNAseq replicates there were additional measures taken to confirm these results.

AUTHORS' RESPONSE: Thank you. As suggested, we have clarified this point in the revised manuscript:

“Although this transcriptome analysis did not include sequenced biological replicates, the high-level expression of PTCDS in the venom gland was validated by subsequent qRT-PCR experiments performed with three independent biological replicates.” (lines 467-470)

Additionally, we have also revised the text to slightly tone down the current conclusions. The changes include:

1. Abstract

Original: “We experimentally demonstrated that PTCDS contributes to the secretion of venom from venom gland cells and is linked to ensure the appropriate storage amount of venom in the venom reservoir of parasitoid wasps.”

Revised: “We experimentally demonstrated that PTCDS is linked to ensure the appropriate storage amount of venom in the venom reservoir of parasitoid wasps.” (lines 29-30)

2. Results Section Headings

Original: “The horizontally acquired PTCDS governs venom protein secretion in venom gland cells”

Revised: “The horizontally acquired PTCDS likely regulates venom protein secretion in venom gland cells” (lines 244-245)

Original: “The eukaryotic-type CDP-diacylglycerol synthase gene is not required for venom protein secretion in parasitoid wasps”

Revised: “The eukaryotic-type CDP-diacylglycerol synthase gene does not regulate stored venom protein amount” (lines 320-321)

3. Results Section content

Original: “These results show that PTCDS regulates venom secretion into the venom reservoir by modulating some protein transport-related genes, including p24-1, Sec24AB, AP-1 μ , CHOp24, and CG1116.”

Revised: “These results show that PTCDS might regulate venom secretion into the venom reservoir by modulating some protein transport-related genes in venom gland, including p24-1, Sec24AB, AP-1 μ , CHOp24, and CG1116.” (lines 284-287)

Original: “These results revealed that the knockdown of PTCDS led to a low number of IVs that transported venom proteins into the canal through the microvillus, which caused a low number of MSEVs, suggesting that the foreign gene PTCDS governs venom protein secretion in the venom gland of parasitoid wasps.”

Revised: “These results revealed that the knockdown of PTCDS led to a low number of IVs that transported venom proteins into the canal through the microvillus, which caused a low number of MSEVs, suggesting that the foreign gene PTCDS is involved in regulating venom protein secretion in the venom gland of parasitoid wasps.” (lines 314-318)

Referee #2:

The revised version of this manuscript deals with the questions raised by me and, I believe, the other reviewer in a fully satisfactory way.

AUTHORS' RESPONSE: We thank the reviewer for the positive feedback and are glad to hear that our revisions have fully addressed the points raised.

Referee #3:

The authors have addressed my minor comments satisfactorily. Note that I have not assessed whether the authors have adequately addressed the comments of the other two reviewers.

AUTHORS' RESPONSE: We thank the reviewer for the insightful comments, which have significantly improved our manuscript.

Dear Prof. Huang,

I am pleased to inform you that your manuscript has been accepted for publication in the EMBO Journal.

You may qualify for financial assistance for your publication charges - either via a Springer Nature fully open access agreement or an EMBO initiative. Check your eligibility: <https://link.springer.com/journal/44318/how-to-publish-with-us>

Yours sincerely,

Yehu Moran
Editor
The EMBO Journal

Please note that it is The EMBO Journal policy for the transcript of the editorial process (containing referee reports and your response letters) to be published as an online supplement to each paper. If you should prefer removal of any referee-only figures included in the point-by-point response(s), e.g. because they may still be used for future publication or because they have been reproduced from published work by others, please do let us know immediately via response email.

More information is available here: <https://link.springer.com/partners/embo-press/editorial-policies#Peer%20review>